# Stratospheric aerosol forcing for CMIP7 (part 1): Optical properties for pre-industrial, historical, and scenario simulations (version 2.2.1)

Thomas J. Aubry<sup>1,2</sup>, Matthew Toohey<sup>3</sup>, Sujan Khanal<sup>3</sup>, Man Mei Chim<sup>4</sup>, Magali Verkerk<sup>1,2</sup>, Ben Johnson<sup>5</sup>, Anja Schmidt<sup>6,7</sup>, Mahesh Kovilakam<sup>8,9</sup>, Michael Sigl<sup>10</sup>, Zebedee Nicholls<sup>11,12,13</sup>, Larry Thomason<sup>8\*</sup>, Vaishali Naik<sup>14</sup>, Landon Rieger<sup>9</sup>, Dominik Stiller<sup>15</sup>, Elisa Ziegler<sup>16</sup>, Isabel Smith<sup>1,2</sup>.

Correspondence to: Thomas J. Aubry (thomas.aubry@earth.ox.ac.uk)

<sup>&</sup>lt;sup>1</sup>Department of Earth and Environmental Sciences, University of Exeter, Penryn, UK

<sup>&</sup>lt;sup>2</sup>Now at: Department of Earth Sciences, University of Oxford, Oxford, UK

<sup>3</sup> Institute of Space and Atmospheric Studies, University of Saskatchewan, Saskatoon, S7N 5A2, Canada

<sup>&</sup>lt;sup>4</sup>Department of Mathematics and Statistics, University of Exeter, Exeter, UK

<sup>10 5</sup>Met Office Hadley Centre, Exeter, UK

<sup>&</sup>lt;sup>6</sup>Institute of Atmospheric Physics (IPA), German Aerospace Center (DLR), Oberpfaffenhofen, Germany

<sup>&</sup>lt;sup>7</sup>Meteorological Institute, Ludwig Maximilian University of Munich, Munich, Germany

<sup>&</sup>lt;sup>8</sup>NASA Langley Research Center, Hampton Virginia USA

<sup>&</sup>lt;sup>9</sup>Environment and Climate Change Canada, Victoria BC, CA

<sup>15 10</sup> Institute of History & Oeschger Centre for Climate Change Research, University of Bern, Bern, Switzerland

<sup>&</sup>lt;sup>11</sup>Climate Resource, Berlin, Germany

<sup>&</sup>lt;sup>12</sup>Energy, Climate and Environment Program, International Institute for Applied Systems Analysis (IIASA), 2361 Laxenburg, Austria

<sup>&</sup>lt;sup>13</sup>School of Geography, Earth and Atmospheric Sciences, The University of Melbourne, Melbourne, Victoria, Australia

<sup>20 &</sup>lt;sup>14</sup>Geophysical Fluid Dynamics Laboratory, Princeton NJ, USA

<sup>&</sup>lt;sup>15</sup>Department of Atmospheric and Climate Science, University of Washington, Seattle WA, USA

<sup>&</sup>lt;sup>16</sup>Department of Geosciences, University of Tübingen, Tübingen, Germany

<sup>\*</sup>Now retired

#### Abstract.

Stratospheric aerosols, most of which originate from explosive volcanic sulfur emissions into the stratosphere, are a key natural driver of climate variability. They are thus a forcing provided by the Coupled Model Intercomparison Project (CMIP) Climate Forcings Task Team to climate modelling groups participating in phase 7 of CMIP. For the historical period, we provide two datasets covering 1750-2023: i) a volcanic upper tropospheric-stratospheric sulfur emission dataset, documented in a companion paper; and ii) a stratospheric sulfate aerosol optical property dataset, which we document here at version 2.2.1. For the satellite era (from 1979 onwards), stratospheric aerosol optical properties are derived from the Global Space-based Stratospheric Aerosol Climatology (GloSSAC) dataset. For the pre-satellite era (1750-1978), optical properties are derived from our volcanic SO<sub>2</sub> emission dataset using a new version of the reduced-complexity volcanic aerosol model Easy Volcanic Aerosol (Height) (EVA H). A background, non-volcanic stratospheric aerosol climatology is derived from the 1998-2001 period with a trend over 1850-1978 accounting for increasing anthropogenic aerosols. A monthly stratospheric aerosol climatology is derived from the 1850-2021 average for both pre-industrial and Scenario (future) simulations, with a 9-year ramp over 2022-2030 for scenario simulations to ensure a smooth transition from the historical period. Our methodology to produce historical aerosol optical properties significantly differs from CMIP6 for the pre-satellite era, and the resulting forcings in turn largely differ. In particular, the CMIP6 dataset was mostly based on the sparse and uncertain pyrheliometer record, which resulted in strongly underrepresented emissions from small-to-moderate magnitude eruptions. The resulting bias is addressed in CMIP7, which is entirely emission-derived in the pre-satellite era and uses more recent ice-core-based volcanic sulfur emission inventories than CMIP6. Our approach results in an overall larger volcanic aerosol forcing for CMIP7, with the 1850-2014 mean mid-visible global mean stratospheric aerosol optical depth (SAOD) in CMIP7 (0.0138) being 29% higher than in CMIP6 (0.0107). The pre-industrial mean of the same variable is 26% higher in CMIP7 (0.0135, derived from the historical 1850-2021) than CMIP6 (0.0107, derived from the historical 1850-2014 mean). Using a reduced-complexity climate model, we simulate a global mean surface temperature that is 0.07°C colder for 1850-1900 when using the CMIP7 dataset 50 instead of CMIP6, whereas 2000-2014 is 0.03°C warmer in CMIP7. Our dataset also exhibits lower forcing for 1960-1980, resulting in temperatures 0.06°C warmer when averaged over 1960-1990, a period for which CMIP6 climate models exhibit a cold bias. Given the large uncertainties characterizing the dataset, in particular for the pre-satellite era, we advise against treating the CMIP7 or CMIP6 dataset as uniquely superior for any specific year and highlight the need for further evaluation. We conclude the study by discussing sources of uncertainty for the dataset, future research avenues to improve it, as well as requirements to operationalize the production of the dataset, i.e. extend it and update it on an annual basis instead of every 5-7 years following CMIP cycles.

#### 1 Introduction


## 1.1 Climate forcings for the CMIP7 Assessment Fast Track

The Assessment Fast Track (AFT) of Phase 7 of the Coupled Model Intercomparison Project (CMIP7) is delivering a set of key climate model experiments across all participating modelling centers, including pre-industrial control simulations, historical (1850-2021) simulations and scenario simulations (2022 to 2100 and beyond) (Dunne et al., 2025). Key to these experiments is the provision of forcing datasets used consistently by all participating models. To this end, a CMIP Climate Forcing Task Team (https://wcrp-cmip.org/cmip7-task-teams/forcings/) was formed to provide key forcings, i.e. external drivers of climate change prescribed in climate model simulations. In addition to forcing provision, the documentation of forcing datasets and progress towards operationalization of forcing production (Naik et al., 2025) are key objectives of the task team. Among the forcings provided, stratospheric aerosol represents one of the most important natural drivers of climate variability (e.g. Santer et al., 2015; Sigl et al., 2015). Sulfate aerosols formed from sulfur gases injected by high-intensity volcanic eruptions are the main contributor to stratospheric aerosols (Kremser et al., 2016). Other contributions include injections by pyrocumulonimbus (pyroCb, e.g. Peterson et al., 2021; Damany-Pearce et al., 2022), aerosols from meteorites (e.g. Schneider et al., 2021), and transport of aerosol or aerosol precursors from the troposphere, some of which originate from anthropogenic emissions (e.g. Hannigan et al., 2022; Brodowsky et al., 2024). Stratospheric aerosols originating from spacecrafts have recently been measured in-situ and their contribution to stratospheric aerosol loading is expected to increase in the future (Murphy et al., 2023).

# 75 1.2 Overview of the CMIP6 stratospheric aerosol optical property dataset

For Phase 6 of CMIP (CMIP6), historical stratospheric aerosol forcing consisted of a set of aerosol properties (Luo et al., 2018; Fig. 1). Unlike CMIP5, CMIP6 was the first time that a consistent dataset was provided to all modelling centers for stratospheric aerosol forcing, representing a major undertaking. The CMIP6 team also generated datasets on a bespoke wavelength grid for each model. No peer-reviewed documentation of the dataset is available to our knowledge, and the following description represents our best understanding of the CMIP6 dataset based on the existing brief documentation (Luo et al., 2018) and data they provided. For the satellite era (from 1979), aerosol optical properties were derived from the Global Space-based Stratospheric Aerosol Climatology version 1.1 (GloSSAC, Thomason et al., 2018), which provides extinction at two wavelengths (525 and 1020 nm) (Fig 1.a). Before the satellite era, aerosol optical properties were derived from a combination of the following (Fig 1.a):

i. For seven large eruptions, they were derived from an ice-core-based volcanic emission inventory (Gao et al., 2008) using the AER2D aerosol model (Arfeuille et al., 2014) (Fig 1.c). The considered eruptions are an eruption in the early 1860s with uncertain attribution, Krakatau (1883), Tarawera (1886), Santa Maria (1902), Katmai (1912), Agung





(1963), and Fuego (1974). Some of the simulations, or injection parameters used, might have been rescaled to better match available pyrheliometer measurements (Stothers, 1996).

- ii. Outside of the periods associated with the above seven eruptions, the CMIP6 documentation states that pyrheliometer data from Stothers (1996) were used to reconstruct "minor" volcanic perturbations if a significant perturbation was detected in these instruments. From Stothers (1996), we understand that over 1881-1960, this approach was used for a total of 97 months distributed within 11 years. To obtain a stratospheric aerosol optical depth (SAOD) at 550 nm, the pyrheliometer optical depth was scaled by an unspecified size dependent conversion factor taking values typically between 1.2-1.5. SAOD at the station(s) location(s) are then used to derive SAOD perturbation over 90°S-0°S, 0°N-90°N or 30°N-90°N (Fig 1.b and d). Based on Luo et al. (2018) and Stothers (1996), we understand that for 87% of the months where pyrheliometer data was used, data from a single station was available. Although the CMIP6 documentation only mentions "pyrheliometer" or "photometer" data and cites Stothers (1996), this paper only covers 1881-1960. 1850-1881 SAOD perturbations that are not model-derived might have been derived from lunar eclipse measurements (Stothers, 2007). 1961-1978 SAOD perturbations that are not model-derived might have been derived from the pyrheliometer measurements compiled in Stothers (2001).
- iii. For any other time period (i.e. pre-1979 and not covered by i or ii), a climatology derived from the volcanically quiescent 1999-2002 period was imposed, with a linear trend between 1850 and 1978 to represent increasing anthropogenic tropospheric aerosol emissions, some of which is transported into the stratosphere. The climatology was added to volcanic perturbations assessed for large (i, emission and model derived) and small (ii, mostly pyrheliometer derived) eruptions.

No matter the source dataset used in CMIP6, the REMAPv1 algorithm (Jörimann et al., 2025) was used to convert available data to full aerosol optical properties (e.g. extinction coefficient, asymmetry factor, or surface area density) at wavelengths requested by each modelling center (for wavelength-dependent variable). The pre-industrial stratospheric aerosol forcing (Eyring et al., 2016), also imposed in Scenario simulations (O'Neill et al, 2016), was taken as the average of the historical dataset, i.e. 1850-2014 average.




Figure 1: a) CMIP6 global mean SAOD at 550nm. Shaded labelled periods show where the data is primarily derived from pyrheliometer measurements (blue shading, labelled "pyr" or "pyr?" if CMIP6 reports pyrheliometer use but we suspect a different data source), model simulations (yellow shading, labelled "mod") or satellite measurements (purple shading, labelled "sat"). For any other period, there is no volcanic SAOD perturbation. Panels b-e show the latitudinal SAOD distribution for four selected 3-year periods, illustrating examples for which the data was primarily derived from pyrheliometer measurements, model simulations or satellite measurements.

The CMIP6 dataset is characterized by long periods with no perturbation in stratospheric aerosol optical properties prior to the satellite era. Over 1850-1978, we estimate that 78 years have no perturbation, with the longest period with no perturbation lasting 28 years, and seven periods with no perturbation that are at least 5-year long. The satellite record evidences that small-to-moderate magnitude eruptions (defined here as injecting less than 3 Tg SO<sub>2</sub>) injecting sulfur into the stratosphere in fact typically occur multiple times per year (e.g. Carn et al., 2016; Schmidt et al., 2018; Chim et al., 2023). In addition to contributing to the mean stratospheric aerosol forcing, these eruptions also contribute to decadal climate variability (Santer et al., 2015). For example, the 1998-2007 period was relatively quiescent, whereas the 2008-2017 period saw several small-to-moderate magnitude eruptions contributing a radiative forcing of -0.1 W/m² (Schmidt et al., 2018). The lack of representation of small-to-moderate magnitude eruptions for the pre-satellite era in CMIP6 is likely caused by the difficulty of detecting these eruptions in the sparse and uncertain pyrheliometer and lunar eclipse record. Recently, Chim et al. (2023, 2025) demonstrated that regardless of the level of volcanic activity for the rest of the century, the CMIP6 ScenarioMIP 2015-2100 mean stratospheric aerosol forcing is very likely to be exceeded. The primary reason is that both the CMIP6 pre-industrial (Eyring et al., 2016) and 2015-2100 Scenario (O'Neill et al., 2016) stratospheric aerosol forcing are the 1850-2014 average of the



historical forcing, and that this average value largely does not account for small-to-moderate magnitude eruptions before 1979. Such bias would thus affect Scenario simulations, pre-industrial control simulations as well as forcing mean level and trends in historical simulations.

Whereas the CMIP6 historical stratospheric aerosol forcing dataset was generated from a range of methods, for some CMIP6 community MIPs, stratospheric aerosol forcing was derived from volcanic emissions using the reduced-complexity aerosol model Easy Volcanic Aerosol model (EVA, Toohey et al., 2016). This includes the Paleoclimate Model Intercomparison Project last millennium experiment (Jungclaus et al., 2017) for which the recommended volcanic forcing dataset was EVA-140 eVolv2k (Toohey and Sigl, 2017), as well as experiments from the Volcanic forcing model intercomparison project (VolMIP, Zanchettin et al., 2016), such as volc-long and volc-cluster experiment series. One important technical evolution since the creation of the CMIP6 dataset is the increasing number of climate models capable of interactively simulating stratospheric aerosol from precursor emissions, including those participating to the Interactive Stratospheric Aerosol MIP (ISA-MIP, Timmreck et al., 2018; Clyne et al., 2021; Quaglia et al., 2023) and more recent additions (e.g. Gao et al., 2023; Ke et al., 145 2025). These models push the frontiers of our understanding of stratospheric aerosol processes and are critical to understand volcanic impacts on climate (e.g. Marshall et al., 2022), the potential forcing and impacts of climate intervention through stratospheric aerosol injections (e.g. Visioni et al., 2021), and the forcing of increasingly frequent pyrocumulonimbus emissions into the stratosphere (e.g. Yu et al., 2023). No stratospheric aerosol precursor emission dataset was provided for CMIP6, although Neely and Schmidt (2016) curated the VolcanEESM volcanic SO<sub>2</sub> emission dataset, which was used by one modelling group to run historical simulations (Davis et al., 2023). 150

#### 1.3 Overview of the CMIP7 stratospheric aerosol optical property dataset

To make progress on the challenges identified for CMIP6, the CMIP7 stratospheric aerosol dataset was created with the following objectives:

- Provide detailed documentation of the dataset in peer-reviewed publications.
- Cater to the needs of models with interactive stratospheric aerosol models by providing a corresponding volcanic SO<sub>2</sub>
  emission dataset.
  - Ensure consistency between the provided emission and aerosol optical property datasets, as well as between the methodology used to provide our datasets, and that expected to be used in community MIPs.
  - Minimize biases in the representation of small-to-moderate magnitude eruptions in the pre-satellite era.
- Produce the dataset from 1750 to support potential extended historical simulations starting from 1750, as well as simulations from reduced-complexity climate models which commonly start in 1750.





Here we document the aerosol optical property dataset produced for CMIP7 (Figure 2). This includes a brief overview of the CMIP7 upper tropospheric-stratospheric (UTS) volcanic sulfur emission dataset, which will be documented in detail in the companion part 2 paper to be submitted by November 2025 in the same special issue (in the meantime, we refer to Aubry et al., 2025, which briefly documents both CMIP7 datasets). Version 2.2.1 of both datasets are documented, i.e. the version frozen for use in the CMIP7 Assessment Fast Track. The CMIP7 stratospheric aerosol optical property dataset is derived from GloSSAC for 1979-2023 (section 2.2). For 1750-1978, volcanic perturbations are derived from the CMIP7 UTS volcanic sulfur emission dataset (section 2.1 and Aubry et al., 2025) using version 2 of the EVA\_H model (section 3). Consistency between the satellite and pre-satellite era parts of the datasets as well as between the emission and aerosol optical property dataset is maximized by calibrating EVA\_H v2 against the satellite-era portion of both datasets. Section 4 provides details on the production of the dataset including merging of satellite and pre-satellite era products, making of pre-industrial and Scenario datasets, and provision of scripts to interpolate our dataset at arbitrary wavelength. A brief comparison between the CMIP6 and CMIP7 datasets is presented in section 5. Last, section 6 discusses requirements to operationalize the production of our aerosol optical property dataset, as well as key uncertainties and directions for future improvements.

Historical datasets cover 1750-2023 at version 2.2.1. Preindustrial control and Scenario climatologies derived from 1850-2021 average of historical.

Figure 2: Graphical depiction of the CMIP7 stratospheric aerosol forcing datasets, i.e. the upper tropospheric-stratospheric (UTS) volcanic SO<sub>2</sub> emission dataset (see overview in section 2.1, Aubry et al., 2025, and upcoming documentation in a companion paper) and the stratospheric aerosol optical property dataset (documented here). Key source datasets for pre-satellite (1750-1978) and satellite (1979-2023) era emissions as well as satellite-era aerosol optical properties are indicated. Version 2 of the EVA\_H volcanic aerosol model (section 3) used to derive pre-satellite era aerosol optical properties is calibrated against satellite-era emission and optical property datasets.

#### 2 Source datasets

#### 2.1 Overview of the upper tropospheric-stratospheric volcanic SO<sub>2</sub> emission dataset

Version 2.2.1 of our emission dataset (Figure 3) is documented in a companion paper (Aubry et al., 2025) and only briefly summarized here. In short, it provides the SO<sub>2</sub> injection mass, altitude, location (latitude and longitude) and date for all





eruptions that led to potential stratospheric aerosol enhancement for 1750-2023 (Fig. 3). These include eruptions with upper-tropospheric volcanic plume heights, although their emissions largely will not contribute to stratospheric aerosol enhancement in the CMIP7 stratospheric aerosol optical property dataset, or in interactive stratospheric aerosol models that will use the CMIP7 emission inventory. For the satellite era (1979-2023), we use the Multi-Satellite Volcanic Sulfur Dioxide L4 Long-Term Global Database (MSVOLSO2L4, Carn, 2022). Prior to the satellite era, we rely on the following emission datasets:

- i. eVolv2k (Toohey and Sigl, 2017) for 1750-1900 and Sigl et al. (2015) for 1901-1978, which are both derived from bipolar ice-core arrays. These datasets capture large-magnitude eruptions and enable one to constrain the date, SO<sub>2</sub> mass and broad latitudinal band of emission (northern hemisphere, tropical and southern hemisphere).
- ii. The D4i high-resolution Greenland core (Fang et al., 2023) for 1750-1900. This dataset captures numerous northern hemisphere and tropical small-to-moderate magnitude eruptions not detected in the above (i) datasets, and enables one to constrain the date and SO<sub>2</sub> mass.
  - iii. The Global Volcanism Program (GVP) Volcanoes of the World database (Global Volcanism Program, 2025), for which we use Volcanic Explosivity Index 4 and 5 eruptions not identified in (i) and (ii). This dataset enables us to account for more small-to-moderate magnitude eruptions, which remain difficult to detect in bipolar ice-core arrays (i) and are still largely underrepresented even in high-time resolution D4i Greenland ice-core (ii), especially for eruptions in the southern hemisphere (expected to not be captured at all in D4i) and tropics. The GVP database provides the exact volcano and, in some instances, detailed information on eruption date or plume altitude, but it does not provide information on SO<sub>2</sub> mass. To minimize reliance on the GVP database, we matched detected volcanic sulfur injection events in (i) and (ii) to eruptions from the GVP whenever a plausible match existed, with more details on matching provided by Aubry et al. (2025). This resulted in 4 VEI 5 eruptions and 64 VEI 4 eruptions added over 1750-1978 from the GVP database. The chosen VEI 5 SO<sub>2</sub> mass for GVP events is 2.78 Tg SO<sub>2</sub>, i.e. the average of the SO<sub>2</sub> mass for all other VEI 5 eruptions in the dataset, where the mass was constrained from satellite or ice-core data. The chosen VEI 4 SO<sub>2</sub> mass for GVP events is 0.08 Tg SO<sub>2</sub>, and it was chosen so that the time-averaged volcanic SAOD over 1750-1978 for small-to-moderate magnitude eruptions is equal to the satellite constrained mean SAOD anomaly (defined the deviation from its minimum) over 1998-2023, a period with no large-magnitude eruptions.

Regardless of the source dataset used for emissions (MSVOLSO2L4, eVolv2k, Sigl et al., 2015, D4i and GVP), numerous injection parameters are not directly constrained (e.g. SO<sub>2</sub> injection altitude or latitude for unattributed ice-core events). We attributed these parameters based on detailed literature review for each eruption or from empirical relationships to be detailed in the (upcoming) accompanying documentation paper specifically focused on this emissions dataset.

Figure 3: CMIP7 upper tropospheric and stratospheric volcanic  $SO_2$  emission dataset (version 2.2.1). Each circle represents an eruptive event shown as a function of latitude and time, with the circle size indicating the mass of  $SO_2$  and the shading the injection height. Dashed lines show the equator and tropic lines. Modified from Aubry et al. (2025).

#### 220 2.2 GloSSAC dataset



The Global Space-based Stratospheric Aerosol Climatology (GloSSAC) was first introduced in 2018 (Thomason et al., 2018) as an extension of the SPARC Assessment of Stratospheric Aerosol Properties (ASAP) (SPARC, 2006). It provides a multisensor record of stratospheric aerosol optical properties from 1979 onward and has been extensively used in modelling (e.g., Timmreck et al., 2018; Aubry et al., 2020; Quaglia et al., 2023) and observational studies (e.g., Stocker et al., 2019). The dataset consists primarily of aerosol extinction coefficients at 525 and 1020 nm, reported on a monthly 5° latitude by 0.5 km altitude grid. GloSSAC includes a zonal mean tropopause height climatology, derived from the SAGE II 1984-2006 period using the MERRA reanalysis (Rienecker et al., 2011) and WMO thermal tropopause height definition (Thomason et al., 2017). CMIP7 datasets used version v2.22 of GloSSAC (https://doi.org/10.5067/GLOSSAC-L3-V2.22). The latest GloSSAC release, version 2.23, spans 1979–2024 (NASA/LARC/SD/ASDC, 2025).

The Stratospheric Aerosol and Gas Experiment (SAGE) series provides the foundation for GloSSAC, with measurements from 1979–1981 (SAGE), 1984–2005 (SAGE II), and 2017–present (SAGE III/ISS). Periods without SAGE coverage are supplemented by complementary spaceborne and ground-based observations (Thomason et al., 2018; Kovilakam et al., 2020), producing a continuous long-term record. Remaining gaps, particularly between December 1981 and September 1984

overlapping the perturbation resulting from the 1982 El Chichón eruption, and in the aftermath of the 1991 Mt. Pinatubo 235 eruption, have been reconstructed from multiple data sources (Thomason et al., 2018; Kovilakam et al., 2020), with ongoing work to improve their representation. Additionally, single-wavelength extinction coefficients from the Optical Spectrograph and Infrared Imaging System (OSIRIS, Rieger et al., 2019) and Cloud Aerosol Lidar and Infrared Pathfinder Satellite Observations (CALIPSO, Kar et al., 2019) were used to bridge post-SAGE II period (August 2005- May 2017), with a conformance procedure to reduce instrument biases (Kovilakam et al., 2020). Current efforts focus on applying reconstruction 240 methods based on Thomason et al. (2021) to improve extinction coefficient estimates during volcanic events in this period, and on evaluating Ozone Mapping and Profiler Suite Limb Profiler (OMPS) aerosol extinction products (Kovilakam et al., 2025) to improve coverage following the termination of CALIPSO and the decline in OSIRIS observations. To produce the CMIP7 aerosol optical property dataset, three pre-processing steps were applied to GloSSAC's extinction at 245 525 and 1020 nm, the two variables we used. First, we masked any data strictly below the altitude of the tropopause height provided in GloSSAC. Second, we corrected a few values of 1020 nm / 525 nm extinction ratios, mostly in the lowermost tropical stratosphere, in the aftermath of the Pinatubo 1991 eruption. Very low values of this extinction ratio, highly localized in periods and areas of the stratosphere with relatively high values, resulted in unrealistically small values of effective radius when processed through our Mie scattering code. We chose to correct any datapoint with 525 nm extinction >0.0028 km<sup>-1</sup> and 250 with 1020 nm / 525 nm extinction ratio <0.25. We left the value of 525 nm extinction unchanged and corrected the 1020 nm extinction value by setting the extinction ratio equal to that of the closest data point with an uncorrected value, where we defined closest as closest in altitude, at the same time and latitude. This resulted in correcting a total of 974 data points (0.1% of the dataset), with 95% of them in the extratropical lowermost stratosphere. Less than 4% of the data was corrected for any given month over 1991-1994, and less than 0.6% of data for the handful of months affected over 2005-2021. Overall, this correction prevents unrealistic values of effective radius, surface area density, volume density and H<sub>2</sub>SO<sub>4</sub> number density whilst having no effect on the mid-visible extinction from GloSSAC. Third, the GloSSAC dataset covers 77.5°S-77.5°N with 5° resolution. To produce a global dataset, we added data at 82.5°S/N and 87.5°S/N by simply assuming the same value of extinction at these latitudes as at 77.5°S/N. This ad-hoc assumption to obtain global coverage affects less than 1.5% of Earth's surface area.

## 3 Volcanic aerosol model (EVA H, v2)



## 3.1 EVA H v1: overview and challenges.

The aerosol model we employ (EVA\_H version 2) is an update to EVA\_H version 1 (Aubry et al., 2020), which was itself an extension of the Easy Volcanic Aerosol (EVA) model, a reduced-complexity volcanic aerosol model developed by Toohey et al. (2016). Using simple parameterizations for stratospheric sulfate aerosol production, loss, effective radius and SAOD, EVA takes as input volcanic SO<sub>2</sub> injection parameters (SO<sub>2</sub> injection mass, latitude and date) to produce 4-dimensional (latitude,







altitude, time and wavelength) aerosol optical properties. In EVA, the stratosphere is divided into three boxes corresponding to three latitudinal bands, and shape functions are used to produce spatially resolved output. EVA was originally calibrated using the Pinatubo 1991 eruption. The model is extremely simple compared to global atmospheric models simulating the chemical, microphysical and transport processes required to simulate the life cycle of stratospheric aerosols, and also much simpler than the AER2D model (Arfeuille et al., 2014) used to generate the forcing associated with a few eruptions in CMIP6 (Figure 1). However, the VolMIP Tambora experiment demonstrated major discrepancies between complex interactive stratospheric aerosol models (Zanchettin et al., 2016; Clyne et al., 2021) and EVA represents a computationally inexpensive, semi empirical middle-ground between them. It has been used in numerous climate modelling applications including CMIP6 VolMIP (Zanchettin et al., 2016), PMIP (Jungclaus et al., 2017), and decadal projections (Sospedra-Alfonso et al., 2024). Employing a low-cost model was also critical in producing the CMIP7 dataset, allowing us to easily revise the numerous dataset versions produced e.g. when revisions to the emission dataset were decided.

From EVA, Aubry et al. (2020) developed the EVA\_H extension, going from a 3-box to an 8-box model by adding three vertical bands corresponding to the lowermost extratropical stratosphere, the lower stratosphere (16-20 km), and the middle stratosphere (20-40 km). With aerosol loss timescales dependent on both latitude and altitude, the simulated evolution of total sulfate aerosol burden and global mean aerosol optical properties is dependent on SO<sub>2</sub> injection latitude and altitude, which was not the case in EVA. The vertical structure of the aerosol optical properties also evolves in time, whereas only the latitudinal structure was evolving in EVA. Last, EVA\_H was calibrated against the 1979-2015 eruption time series using an earlier version of MSVOLSO2L4 (for volcanic SO<sub>2</sub> emissions) and GloSSAC (for stratospheric aerosol optical properties) instead of Pinatubo 1991 only for EVA. Differences between EVA and EVA\_H generated forcing and their implications for simulating the climate response to volcanic eruptions are investigated in Bilbao et al. (2025).

When calibrating EVA\_H v1, one of the main challenges was to implement parameterizations better capturing the forcing evolution for eruptions spanning a broad range of injection mass, latitude, altitude and season. Aerosol loss timescale dependent on altitude and latitude significantly improved the model performance. However, this was not the case for the aerosol production timescale which led to a single, prescribed aerosol production timescale of 7.8 months, whose value was largely driven by the good resulting fit for the Pinatubo 1991 eruption. Despite EVA\_H loss timescales dependence on injection latitude and altitude, the long, prescribed aerosol production timescale limits the sensitivity of the model to eruption characteristics and led to overestimated aerosol lifetime for relatively small eruptions over the 2001-2015 (Aubry et al., 2020). The same issue was evident for the 2019 Raikoke eruption, for which a production timescale of 2.8 months was more adequate (Vernier et al, 2024). The reason for which more sophisticated parameterization of aerosol production timescale did not significantly improve model performance lies in the challenges in reconciling the MSVOLSO2L4 emission inventory and GloSSAC aerosol optical property dataset, which, for more realistic temporal evolution of SAOD perturbations, can lead to worsened model performance. For example, EVA\_H already tends to overestimate the peak SAOD of numerous eruptions over 2001-2015, which could largely be caused by biases in MSVOLSO2L4 or GloSSAC, both for Pinatubo 1991 (the eruption with the most weight in the calibration) and small-to-moderate magnitude eruptions (challenging to simulate with EVA\_H v1







when matching Pinatubo). When decreasing the model aerosol production timescale, peak SAOD becomes even larger which thus decreased the model performance despite a more realistic time-evolution of SAOD perturbation.

In the following, we describe all updates to EVA\_H v1 made to develop EVA\_H v2 used to produce the CMIP7 dataset. Although we strive to provide a basic understanding of EVA\_H v1 in this paper, we refer to Aubry et al. (2020) for an extensive documentation of v1 of the model.

#### 3.2 Updates to aerosol production timescale and model calibration

To improve EVA\_H v1, we introduce a new parameterization of the aerosol production timescale dependent on SO<sub>2</sub> injection parameters. Fitting EVA\_H v1 to single eruptions instead of the full 1979-2023 sequence suggested that the aerosol production timescale was a power-law function of the injected SO<sub>2</sub> mass, and a linear function of the injection height. The latter is consistent with previous findings that the SO<sub>2</sub> e-folding time increases linearly with altitude (Carn et al., 2016). Consequently, we parameterize the aerosol production timescale as:

$$\tau_{prod} = a_1 \left( 1 + a_2 H_{SO2} \right) M_{SO2}^{a_3} \tag{1}$$

where  $a_1$ ,  $a_2$  and  $a_3$  are three model parameters replacing the previous single value of the production timescale, and imposing a minimum value for  $\tau_{prod}$  of 0.1 month.

As in Aubry et al. (2020), we used a genetic algorithm to calibrate EVA\_H v2 using MSVOLSO2L4 as model input, and minimizing the error on the aerosol optical depth (AOD) in each EVA H box when compared to GloSSAC. Whereas calibration of EVA H v1 was against the 1979-2015 period, we used the 1979-2023 period to calibrate EVA H v2, a significant extension of the calibration dataset given the short duration of our satellite record. GloSSAC was pre-processed by removing the 1998-2001 monthly mean climatology of extinction to obtain extinction anomaly, and tropospheric extinction values were masked using the GloSSAC-provided tropopause height (Figure 4.a, black line). MSVOLSO2L4 was preprocessed by grouping together injections from the same volcano and eruption, by summing SO<sub>2</sub> masses, averaging SO<sub>2</sub> injection height weighted by SO<sub>2</sub> mass, and taking the date of the first injection. Additionally, we introduced an ad-hoc SO<sub>2</sub> injection of 0.7 Tg SO<sub>2</sub> at 20 km a.s.l., 33°S in Dec 2019 to represent the Black Summer Australian pyrocumulonimbus. EVA H does not presently have the capability to represent wildfire injections which are primarily organic and black carbon (e.g. Yu et al., 2023). Introducing an ad-hoc SO<sub>2</sub> injection enables us to not bias the model calibration. The absence of significant injection to realistically reproduce the large Southern Hemisphere SAOD perturbation in 2020-2021 (e.g. Kloss et al., 2021) could indeed have been compensated during calibration by biases in values of model parameters such as fast aerosol transport from the tropics to the extra tropics, or long production and/or loss timescales for 2019 eruptions of Raikoke and Ulawun which would have biased values of a<sub>1</sub>-a<sub>3</sub> (Equation 1). A very large uncertainty was used for the SO<sub>2</sub> mass of the Black Summer pyrocumulonimbus to ensure that it has no minimal influence on the calibration. Numerous other pyrocumulonimbus perturbations could have been introduced (e.g. 2017 Canadian pyrocumulonimbus, Kloss et al. 2019, 2021)







but were deemed less risky to bias the model calibration. The processed emission inventory used to calibrate EVA\_H v2 is provided in Supplementary Table 1.

Critically, we modified the calibration process to ensure that it does not favor closest matching exactly GloSSAC aerosol optical properties given specific SO<sub>2</sub> injection estimate from MSVOLSO2L4 at the expense of realistic time evolution of aerosol optical property perturbations. In addition to model parameters, calibration parameters for v2 include the SO<sub>2</sub> mass of all eruptions considered in the calibration. We calculate an uncertainty on injected stratospheric mass by assuming that eruptions other than Pinatubo have a factor 2 uncertainty on mass reported in MSVOLSO2L4, and 4 km uncertainty on plume height. Pinatubo has the most weight in the calibration so no matter what the SO<sub>2</sub> mass used, the resulting model will fit the Pinatubo signal. Consequently, we fix the Pinatubo mass to its best estimate in MSVOLSO2L4 (15 Tg SO<sub>2</sub>), but add an uncertainty of 33% for all other masses (corresponding to an assumed 33% uncertainty on Pinatubo mass). When calibrating the model, we then look for the combination of model parameter values and erupted SO<sub>2</sub> mass, within uncertainty for each eruption, minimizing our error metric. This enables the genetic algorithm to adjust SO<sub>2</sub> masses within a realistic range, facilitating matching of both SAOD perturbation magnitude and time-evolution. To keep the model calibration reasonably quick, we first did an initial model parameter search allowing for adjustment of SO<sub>2</sub> masses for the 44 eruptions (out of 182) with a best estimate of stratospheric SO<sub>2</sub> mass higher than 0.1 Tg SO<sub>2</sub>. We then did a refined search allowing for adjustment of SO<sub>2</sub> masses for the 88 eruptions with an upper end of stratospheric SO<sub>2</sub> mass higher than 0.1 Tg SO<sub>2</sub>.

Table 1 shows all calibrated model parameter values. The resulting global mean SAOD predicted by EVA\_H v2 is shown as a red-dotted line on Figure 4.a, with the red continuous line showing the same model run without adjusted SO<sub>2</sub> masses. When running EVA\_H with MSVOLSO2L4 unadjusted SO<sub>2</sub> masses, both v1 (yellow line on Figure 4.a) and v2 typically overestimate SAOD perturbations associated with 21st century small-to-moderate magnitude eruptions, with v2 typically worst. However, thanks to the implementation of a new parameterization for the production timescale (Equation 1) combined with the new calibration process, EVA\_H v2 clearly captures much better than EVA\_H v1 the rise and decay timescales of SAOD perturbations associated with a broad range of eruptions (Figure 4.b-f). EVA is not shown in these figures, but it would exhibit the same time evolution of global mean SAOD independently from SO<sub>2</sub> injection parameters. Comparing model parameters in EVA\_H v1 vs v2, v1 has an 18% smaller scaling factor to convert the total stratospheric sulfate aerosol burden into global mean SAOD at 525 nm. This difference is largely consistent with the fact that the estimated Pinatubo 1991 SO<sub>2</sub> injection mass decreased by 17%, from 18 to 15 Tg SO<sub>2</sub>, between the MSVOLSO2L4 used to calibrate v1 and v2 versions of EVA\_H. The difference between tropical and extra-tropical aerosol loss timescales is greater in EVA\_H v2 compared to v1, which is an improvement given the lack of sensitivity to eruption latitude of EVA\_H v1 compared to interactive stratospheric aerosol models (Aubry et al., 2020). The aerosol production timescale of 7.8 months in EVA\_H v1 is now Equation 1 with  $\tau_{prod} = 0.012 (1 + 9.8 \, H_{SO2}) \, M_{SO2}^{0.31}$  months. For Pinatubo 1991 (15 Tg SO<sub>2</sub>, 25 km a.s.l.), this results in a production

timescale of 6.8 months. For Soufriere Hills 2006 (0.2 Tg SO<sub>2</sub>, 18 km a.s.l.), this results in a production timescale of 1.3 months. Overall, the production timescale scales linearly with height and increases by a factor of 2 when  $SO_2$  mass increases by a factor of 10. Last, we updated the effective radius calibration (Table 1) in line with both changes to the model equations (this section) and aerosol size distribution and resulting Mie look-up table (section 3.3). We also allow local effective radius as small as 0.06  $\mu$ m instead of 0.1  $\mu$ m in EVA\_H v1, which enables us to better match high 525 nm / 1020 nm extinction ratios from GloSSAC.



| Parameter                                            | EVA_H v1            | EVA_H v2                                 |  |
|------------------------------------------------------|---------------------|------------------------------------------|--|
| Scaling factor between total stratospheric sulfate   | 0.0187              | 0.0229                                   |  |
| aerosol burden and global mean SAOD at 525nm         |                     |                                          |  |
| (A, (Tg S)-1)                                        |                     |                                          |  |
| Aerosol production timescale $(\tau_{prod}, month)$  | 7.8                 | $0.012 (1 + 9.8 H_{SO2}) M_{SO2}^{0.31}$ |  |
| Aerosol loss timescale (extratropical middle         |                     |                                          |  |
| stratosphere) ( $\tau_{loss}^{1,3}$ , month)         | 2.3                 | 1.9                                      |  |
| Aerosol loss timescale (tropical middle              |                     |                                          |  |
| stratosphere) ( $\tau_{loss}^2$ , month)             | 9.5                 | 18.8                                     |  |
| Aerosol loss timescale (extratropical lower          |                     |                                          |  |
| stratosphere) ( $\tau_{loss}^{4,6}$ , month)         | 2.7                 | 2.7                                      |  |
| Aerosol loss timescale (tropical lower stratosphere) |                     |                                          |  |
| $(\tau_{loss}^5, \text{month}), \text{month})$       | 16.1                | 57.7                                     |  |
| Aerosol loss timescale (extratropical lowermost      |                     |                                          |  |
| stratosphere) ( $\tau_{loss}^{7,8}$ , month), month) | 3.8                 | 4.5                                      |  |
| Mixing timescale (\tau_{mix}, month)                 | 10.7                | 9.2                                      |  |
| Global mean effective radius (Reff, µm)              | $0.25M_{SO4}^{1/3}$ | $0.15  M_{SO4}^{1/3}$                    |  |
|                                                      | Minimum local       | Minimum local value of 0.06 μm and       |  |
|                                                      | value of 0.1 μm     | minimum global mean value of 0.115 μm    |  |

Table 1: Parameterization and parameter values used in v1 and v2 of EVA\_H. Symbols in parenthesis refer to symbol used in the EVA\_H v1 documentation paper (Aubry et al., 2020), with superscripts for loss timescales referring to model boxes. For the aerosol production timescale parameterization,  $H_{SO2}$  refers to the SO<sub>2</sub> injection height in km a.s.l., and  $M_{SO2}$  to the mass of SO<sub>2</sub> in Tg S. For the effective radius parameterization,  $M_{SO4}$  refers to the mass of sulfate aerosol in Tg S.



Figure 4: Global mean SAOD time series for GloSSAC, EVA\_H v1 and EVA\_H v2, run using the MSVOLSO2L4 emission inventory modified as specified in section 3.3. In panel a, we additionally show EVA\_H v2 run with adjusted SO<sub>2</sub> masses constrained as part of the calibration process (section 3.3). Panels b-f shows selected time periods with SAOD normalized by its maximum, highlighting how SAOD time evolution compares between models and GloSSAC. Eruptions dominating the sulfur budget for each period and their SO<sub>2</sub> injection parameters are labelled above each panel.

#### 3.3 Updates to aerosol size distribution and Mie look-up tables

As in EVA, the EVA\_H reduced-complexity volcanic aerosol model produces an estimate of the sulfate aerosol mass, effective radius and extinction at 525 nm, then use a lookup table to convert this extinction to other wavelengths and other optical and physical properties. This lookup table is produced using Mie theory, which requires assumptions about the complex refractive index of the aerosol solution, and the aerosol size distribution. For EVA\_H v2, complex refractive index of sulfuric acid solution in water is taken from Biermann et al., (2000), which provides values for a temperature of 215K and a sulfate-water solution with sulfate concentration of 75%. The complex refractive index data was then interpolated to a wavelength grid with 298 data points between 0.15 to 100 µm using the Akima interpolation method (Akima, 1970). The imaginary part of the complex refractive index for wavelengths smaller than 2.65 µm are all set to equal values from Hummel et al. (1988). This value is around 0.004 at 2.5 µm and decreases to near zero at shorter wavelengths indicating negligible absorption by sulfuric acid solution in this part of the spectrum. In comparison, EVA and EVA\_H v1 lookup tables were produced using complex refractive index values from Palmer and Williams (1975) and a coarser spectral resolution complex refractive index with only 29 data points.




Another important update that was made for EVA\_H2 is the use of a more representative aerosol particle size distribution. EVA\_H v1 used a unimodal lognormal distribution with a geometric standard deviation of 1.2. In-situ measurements from ballon-borne measurements have shown that the particle size distribution of volcanic aerosols following the Pinatubo eruption is better represented by a bimodal distribution (Deshler et al., 1993). Furthermore, recent multi-wavelength SAGE III measurements indicate that particle size distributions following small-to-moderate eruptions have a geometric standard deviation around 1.6-2.0 (Wrana et al., 2023). To simultaneously account for eruptions at these different scales, EVA\_H v2 uses a bimodal lognormal distribution with a geometric standard deviation of 1.8 and 1.25 for the smaller and larger sized modes respectively. The bimodal distribution becomes relevant for the EVA\_H v2 in the effective radius range of 0.4 - 0.75 µm, i.e., the bimodal assumption is applicable only for larger eruptions like the El Chichón 1982 and Pinatubo 1991 eruptions, whereas for smaller sized eruptions where aerosol effective radii is below 0.4 µm, it is consistent with a unimodal distribution with a geometric standard deviation of 1.8. The complex refractive index along with the particle size distribution function are then input into the Mie scattering code PyMieScatt (Sumlin et al., 2018) to compute look-up tables of extinction efficiency, single scattering albedo and asymmetry parameter as a function of wavelength and effective radius. Figure 5 shows the resulting dependency of the 1020/525 nm extinction efficiency ratio on effective radius.

Figure 5: 1020:525 nm (GloSSAC wavelengths) extinction efficiency ratio as a function of effective radius for the Mie look-up tables used in EVA\_H v1 and v2. The shading shows the 0.5-99.5 interquantile range of values in GloSSAC.

Beyond optical properties, climate models running with prescribed stratospheric aerosol properties might require the aerosol surface area density, volume density and the H<sub>2</sub>SO<sub>4</sub> number concentration. There properties were not output in v1 of EVA\_H, which we address in v2. The look-up tables described above include values for the volume and surface area density for a nominal sulfate aerosol number concentration of 1 per cubic centimeter. We calculate the sulfate aerosol number concentration as the ratio of the extinction efficiency at 550 nm and the aerosol effective cross section, which is also calculated as part of the look-up table. Last, the H<sub>2</sub>SO<sub>4</sub> number density (nd) is calculated as:

$$nd = \frac{vd \,\rho_a \,N_{av} \,w_a}{M_{H2SO4}} \tag{2}$$

where vd is the aerosol volume density,  $w_a$ =0.75 is the assumed fraction of sulfate in the aerosol,  $\rho_a$  is the aerosol density,  $N_{av}$  the Avogadro number, and  $M_{H2SO4}$  the molar concentration of  $H_2SO_4$ . The aerosol density is calculated following Sandvik et al. (2019) as:

$$\rho_a = (-0.4845 - 0.7074 w_a) T_a + 1186.1 + 621.4 w_a + 573.54 w_a^2$$
 (3)

where  $T_a = 215$  K is the assumed temperature, and  $\rho_a$  is expressed in kg m<sup>-3</sup>.

#### 4 Stratospheric aerosol optical properties for pre-industrial, historical (1750-2023) and scenario simulations

### 4.1 Non-volcanic background stratospheric aerosol optical properties

Section 3 introduces the model used to derive volcanic aerosol optical property perturbation during the pre-satellite era. 430 However, stratospheric aerosol optical properties also include a background, non-volcanic component (Kremser et al., 2016). Note that "non-volcanic" refers to the fact that these aerosols did not originate from eruptions injecting sulfur directly into the stratosphere, nor from eruptions that that injected sulfur into the upper troposphere that was rapidly (typically within weeks) into the stratosphere, e.g. through radiative lofting (e.g. Muser et al., 2020) or Monsoon circulation (e.g. Bourassa et al., 2012). Some sulfur originating from tropospheric volcanic plumes, along with tropospheric sulfur originating from other sources (e.g. anthropogenic), might still be slowly transported into the stratosphere and contribute to this "non-volcanic" background. The 435 non-volcanic background is directly measured for the satellite era in GloSSAC, but must be accounted for during the presatellite era. The late 1990s/early 2000s are commonly used to derive a relatively volcanically-quiescent background stratospheric aerosol optical property climatology owing to the few and small eruptions detected during that time period (e.g. Carn et al., 2016; Schmidt et al., 2018). In CMIP6, 1999-2002 was used to derive a non-volcanic climatology. For CMIP7, we make a slightly different choice of using 1998-2001. We exclude year 2002 because three eruptions injecting nearly 0.1 Tg 440 SO<sub>2</sub> into the stratosphere occur near the end of that year (Ruang in September, Etna in October and Reventador in November),


and these eruptions are followed by a small but clear increase in tropical and NH SAOD in late 2002 (Figure 6.a). We also include year 1998 because no eruption injecting more than 0.01 Tg SO<sub>2</sub> occured over 1996-1998 and although a weak SAOD decay trend is still visible over 1996-1997 as a consequence of the 1991 Pinatubo eruption, it is strongly dampened by 1998 (Figure 6.a). Both for extinction at 525 and 1020 nm, monthly mean volcanically-quiescent climatologies were thus calculated from the 1998-2001 average. Figure 6.b shows the corresponding climatology of 525 nm SAOD, with expected lower SAOD over the tropics, and a seasonal cycle mostly characterized by an increase in SAOD at northern high latitudes over April-July and an increase in SAOD at southern high latitudes over October-January.

In EVA\_H v1, non-volcanic background aerosol optical properties were generated by injecting a steady SO<sub>2</sub> flux in each of the model boxes, with these fluxes calibrated to best match the background aerosol optical properties. This approach resulted in the absence of seasonal cycle, and also caused the background aerosol spatial distribution to be determined by the model shape functions. This approach would have resulted in inconsistencies for the non-volcanic background between the presatellite and satellite era part of the dataset, which was flagged as a major concern by the group within the CMIP Climate Forcing Task Team in charge of producing the ozone forcing for CMIP7 (Michaela Hegglin, personal communication, 2025). Consequently, for EVA\_H v2, to produce full stratospheric aerosol optical property fields, we directly add the GloSSAC-derived 1998-2001 climatologies in extinction at 525 and 1020nm (Figure 6.b) to volcanic perturbations in extinction generated by running the model from volcanic emissions (section 3).

Last, direct observations show an increase in stratospheric carbonyl sulfide (OCS), one of the most important non-volcanic 460 sources of stratospheric sulfur, over 1986-2016, driven by anthropogenic sulfur emissions (Hannigan et al., 2022). This is consistent with model simulations showing an increasing trend in non-volcanic background stratospheric aerosol throughout the historical period (Luo et al., 2018, based on a personal communication by Mike Mills; Davis et al., 2023). In CMIP6 (Luo et al., 2018), based on Sheng et al. (2015), it was estimated that the total background stratospheric sulfur burden was 0.074 Tg S in 1850 and 0.11 Tg S for modern times. Luo et al. (2018) assumed a linear trend in background extinction proportional to 465 these burdens to scale the GloSSAC derived background aerosol optical properties back in time. We follow the same assumption to produce the CMIP7 dataset, i.e. we assume that in year 2000, the background extinction at 525 and 1020 nm is the 1998-2001 mean climatology (Figure 6.a), and that in year 1850, it was 0.074/0.11=67.5% of the 2000 values. We linearly scale background extinction between these two years, and assume a steady climatology before 1850 and after 2000. Although 470 the resulting non-volcanic stratospheric aerosol background has an increasing trend consistent with limited observations and model simulations, it might remain too simplistic. The prescribed trend in CMIP7 background SAOD over the historical period is half the magnitude of the trend in interactive stratospheric aerosol simulations from the UK Earth System Model (UKESM) (Figure 6.c). The UKESM simulations also imply that most of the change occurs between approximately 1950 and 1978. After 1978, when we use directly observed stratospheric aerosol optical properties in CMIP7, UKESM shows a slow-down and 475 plateau of SAOD increase, followed by an apparent decrease from around 2008.


Figure 6: a) SAOD (left axis, dotted lines) and stratospheric volcanic SO<sub>2</sub> emissions (right axis, bars) for the relatively volcanically quiescent 1996-2005 period. Both SAOD and emissions are split over SH (90-25°S), tropical (25°S-25°N) and NH (25-90°N). b) 1998-2001 (shaded in grey on panel a) climatological SAOD at 525 nm. c) Annual global mean background SAOD at 550 nm in CMIP7 (shown for 1820-1978) and as simulated by UKESM v1.0 (Sellar et al., 2019) using CMIP6 emissions and forcings. The UKESM background SAOD was obtained by running historical simulations with interactive stratospheric aerosols, but with no volcanic SO<sub>2</sub> emissions (unpublished to date).

## 4.2 Production of historical 525 nm and 1020 nm extinction

Extinction at 525 nm and 1020 nm are the key variables on which production of the CMIP7 aerosol optical property dataset relies. To produce these variables for 1750-1981, we summed the non-volcanic background (section 4.1) and volcanic



perturbations obtained by running EVA\_H v2 (section 3) using the CMIP7 volcanic SO<sub>2</sub> emission inventory (section 2.1 and Aubry et al., 2025). One modification was made to the emission inventory before running EVA\_H. We injected SO<sub>2</sub> associated with the Agung 1963 eruptions at 27°S (Figure 7.b) instead of the actual latitude of 8.3°S (Figure 7.a). This ensures that most of the Agung aerosol is in the Southern Hemisphere and tropics, instead of a hemispherically symmetric spreading, which results in a better agreement with pyrheliometer measurements (Stothers et al., 2001) and the CMIP6 dataset (Figure 7.c, SO<sub>2</sub> injected between 0-15°S).

Figure 7: SAOD (550 nm) for the 1963-1966 period, mostly characterized by the Agung 1963 eruptions. Panel b and c respectively show the CMIP7 and CMIP6 datasets, whereas panel a shows what the CMIP7 dataset would have been if we had not used a modified latitude for Agung SO<sub>2</sub> injection (27°S instead of 8.3°S) to ensure a stronger Southern Hemisphere forcing.

To produce the full (1750-2023) extinction dataset, we simply merge the 1979-2023 extinction dataset from GloSSAC (section 2.2 and Kovilakam et al., 2020) with the 1750-1981 emission-derived extinction dataset. To harmonize both datasets and avoid a small but steep jump in extinction values at the start of the GloSSAC period (January 1979), for 1979-1981, extinction coefficients are a weighted average of the emission-derived (using EVA\_H) and GloSSAC extinction coefficients, with the GloSSAC weight linearly increasing from 0 in Jan 1979 to 1 in Dec 1981 (Figure 8). The start of the period is imposed by the




start of the GloSSAC dataset, whereas we chose Dec 1981 to avoid merging over a period with large SAOD perturbations associated with the El Chichón eruptions in 1982. Over 1979-1981, GloSSAC and the emission-derived extinction are in good agreement in terms of latitudinal distribution (Figure 8.c-f). However, both at 525 (Figure 8.a) and 1020 nm (Figure 8.b), the emission derived-dataset has  $\approx$ 25-30% smaller SAOD values over the volcanically quiescent 1979 year. This discrepancy comes from differences in GloSSAC between SAOD values for volcanically quiescent year 1979 and years 1998-2001, our chosen period to define the non-volcanic background. The differences might either be caused by trends in background stratospheric aerosols not matching the assumptions we made (section 4.1 and Figure 6), and/or by inconsistencies or quality issues in the GloSSAC dataset before and after 1984, from when SAGE II measurements are used (Kovilakam et al., 2020). Furthermore, for small-to-moderate magnitude eruptions in 1980 (including Mt St Helens,  $\approx$ 0.9 Tg SO<sub>2</sub> at 27 km and 46°N, and Ulawun,  $\approx$ 0.2 Tg SO<sub>2</sub> at 20 km and 5°S) and 1981 (including Alaid,  $\approx$ 1.1 Tg SO<sub>2</sub> at 15 km and 51°N and Pagan,  $\approx$ 0.3 Tg SO<sub>2</sub> at 20 km and 18°N), the emission-derived SAOD perturbations are greater than GloSSAC. This could be caused by errors in GloSSAC, in the emission dataset, or/and biases in EVA\_H. Altogether, although these differences are relatively small in absolute term (

Figure 8: Merging of CMIP7 pre-satellite (EVA\_H run from CMIP7 emissions) and satellite (GloSSAC) era datasets over 1979-1981, illustrated for SAOD at 525 nm (left) and 1020 nm (right). The top panels (a,b) show the global mean SAOD, whereas bottom panels (c-h) show the SAOD.

## 4.3 Production of historical stratospheric aerosol optical properties

| Name | Full name                   | Unit                             | dimensions                           |
|------|-----------------------------|----------------------------------|--------------------------------------|
| ext  | Extinction coefficient      | m <sup>-1</sup>                  | Time, latitude, altitude, wavelength |
| ssa  | Single scattering albedo    | -                                | Time, latitude, altitude, wavelength |
| asy  | Scattering asymmetry factor | -                                | Time, latitude, altitude, wavelength |
| reff | Effective radius            | m                                | Time, latitude, altitude             |
| sad  | Surface area density        | μm <sup>2</sup> cm <sup>-3</sup> | Time, latitude, altitude             |


| vd | Volume density                                | μm <sup>3</sup> cm <sup>-3</sup>                            | Time, latitude, altitude |
|----|-----------------------------------------------|-------------------------------------------------------------|--------------------------|
| nd | H <sub>2</sub> SO <sub>4</sub> number density | Molecule<br>H <sub>2</sub> SO <sub>4</sub> cm <sup>-3</sup> | Time, latitude, altitude |

Table 2: Summary of variables provided in the CMIP7 stratospheric aerosol optical property dataset at version 2.2.1.

Table 2 shows the variables provided in the CMIP7 aerosol optical property dataset at version 2.2.1. All variables are provided as zonal averages. Each variable has dimensions of time, latitude and height, with ext, ssa and asy additionally depending on wavelength. Time ranges from January 1750 to December 2023 with monthly resolution. Latitude ranges from -87.5 to 87.5 degree North with resolution of 5 degrees. Height ranges from 5 to 39.5 km a.s.l. with a resolution of 0.5 km. We provide ext, ssa and asy at the following 39 wavelengths (in  $\mu$ m): 0.16, 0.23, 0.3, 0.39, 0.46, 0.525, 0.53, 0.55, 0.61, 0.7, 0.8, 0.9, 1.01, 1.02, 1.27, 1.46, 1.78, 2.05, 2.33, 2.79, 3.418, 4.016, 4.319, 4.618, 5.154, 6.097, 6.8, 7.782, 8.02, 8.849, 9.708, 11.111, 13.157, 15.037, 17.699, 20.0, 23.529, 35, 50, 75, and 100. This list includes:

- i) wavelengths required by the Rapid Radiative Transfer Model commonly used in multiple climate models, e.g., by EC-Earth (Serva et al., 2024).
  - ii) wavelengths that are key to building the dataset, i.e., GloSSAC wavelengths (0.525 and 1.02  $\mu$ m) and 0.550 used by the reduced-complexity aerosol model EVA\_H v2.
  - iii) Additional wavelengths chosen to have a relatively regularly spaced (in logarithmic space) set of wavelengths.

To produce the full aerosol optical property dataset (Table 2, with wavelengths listed above) from the extinction at 525 and 1020 nm dataset (section 4.2), we:

- 1) Derived the effective radius from the ratio of 525 to 1020 nm extinction coefficient using the EVA\_H v2 Mie look-up table (section 3.3). As in EVA H v2, we impose a minimum local effective radius of 0.06 μm.
- 2) Derived all other properties at arbitrary wavelength from the extinction at 525 nm and the effective radius using the same look-up tables.

Select examples of other variables are compared with CMIP6 in section 5.

## 4.4 Production of pre-industrial and scenario aerosol optical properties

As in CMIP6 (Eyring et al., 2016), the pre-industrial control stratospheric aerosol optical property climatology is chosen to be the monthly mean climatology averaged over the period covered by historical simulations, i.e. 1850-2021 (Dunne et al., 2025). For each variable provided (Table 2), we thus provide a 12-month climatology directly calculated from the 1850-2021 average of the corresponding variable in the historical dataset. Also as in CMIP6 (O'Neill et al., 2016), the ScenarioMIP stratospheric aerosol optical properties are the same as in pre-industrial simulations, with a 9-year linear ramp (instead of 10 in CMIP6 ScenarioMIP) to that climatology from the end of the historical period (i.e. December 2021) (Figure 9). Thus, the Scenario files start in January 2022 even though historical datasets go up to December 2023 at version 2.2.1 and source datasets for

extension to December 2024 are already available. In particular, over 2022-2023, Scenario forcings will not include the stratospheric aerosol forcing associated with the Hunga Tonga Hunga Ha'apai 2022 eruption, characterized by a global mean SAOD perturbation of ≈0.008 (Figure 9), around 15 times less than the Pinatubo 1991 eruption. In addition to the stratospheric aerosol net negative radiative forcing, the Hunga eruption injected 150 Tg of water in the stratosphere (Millán et al., 2022), which was associated with a positive radiative forcing. Although our Scenario dataset and ScenarioMIP forcing will not account for this eruption, the surface temperature impacts of both the aerosol and water vapor forcing are expected to be negligible, with the combined impact uncertain to be a net surface warming or cooling (e.g. Jenkins et al., 2023, Schoeberl et al., 2024). Last, note that as in CMIP6, pre-industrial experiments and any ScenarioMIP experiment will have the same prescribed stratospheric aerosol forcing. However, both the non-volcanic stratospheric aerosol background (Figure 9, UKESM simulations from Chim et al., 2023) and volcanic forcing (Aubry et al., 2021a) are dependent on other forcings and/or the background climate state. Further work is required to constrain these dependencies and provide modelling groups with stratospheric aerosol dataset accounting for these feedbacks in future CMIP versions. Models with interactive stratospheric aerosol capabilities, which can perform emissions-driven runs consistently thanks to the provision of a dedicated volcanic SO<sub>2</sub> emission dataset (section 2.1 and Aubry et al., 2025), will account for these feedbacks.

Figure 9: Transition from historical (blue) to Scenario (yellow) SAOD. From 2033 onwards, the Scenario optical properties are also the same as the pre-industrial ones. Dotted lines show the non-volcanic stratospheric aerosol background as simulated by UKESM v1.1 (with interactive stratospheric aerosol, Chim et al., 2023) using CMIP6 SSP3-7.0 (red), SSP1-2.6 (green) and preindustrial control (black, arbitrary time axis) forcings.

# 4.5 Production of dataset on bespoke wavelength grid

To facilitate use of our dataset in any radiative model, we provide the community with scripts that can be used to interpolate the files we provide on ESGF on any list of wavelength ranges inputted by the user. Optical property, p, at band i is computed from the weighted integral over the band wavelength range  $\lambda_0$  to  $\lambda_1$  as

$$p_i = \frac{\int_{\lambda_0}^{\lambda_1} B_{\lambda}(\lambda) w(\lambda) p(\lambda) d\lambda}{\int_{\lambda_0}^{\lambda_1} B_{\lambda}(\lambda) w(\lambda) d\lambda} \tag{4}$$

where  $B_{\lambda}$  is the spectral irradiance, and w is a property dependent weight. For extinction, w=1, for the single scattering albedo, w is the extinction, and for the asymmetry factor, w is the extinction multiplied by the single scattering albedo. This is implemented by assuming a Planck function for the irradiance and linearly interpolating the original dataset onto a higher-






resolution grid that is used for numerical summation. A simpler, less costly, linear interpolation method using band midpoints is also provided, but this is less accurate, particularly over broad bands, and therefore not generally recommended. All scripts are accessible at <a href="https://github.com/MetOffice/CMIP7">https://github.com/MetOffice/CMIP7</a> volcanic aerosol forcing/. Modelling groups tweaking these scripts or using a different approach should simply document it.

## 5 Comparison with CMIP6

## 5.1 Historical (1750-2023) 550nm stratospheric aerosol optical depth

Figure 10 compares CMIP6 and CMIP7 global mean 550nm SAOD time series, focusing on 1850 onwards. The two datasets are unsurprisingly in close agreement for the satellite era, a period for which they both used GloSSAC. Minor disagreements are apparant over: i) 1979-1982, the period over which our dataset is a weighted average of GloSSAC and our emission-derived dataset (Section 4.2, Figure 8); ii) the Pinatubo 1991 eruption, with peak SAOD roughly 10% higher in CMIP7 owing to GloSSAC revisions (Aubry et al., 2020; Kovilakam et al., 2020); iii) the El Chichón 1982 eruption, with peak SAOD roughly 15% higher in CMIP6 due to revisions in the handling of airborne lidar data used in the tropics in GloSSAC; iv) the 2005-2017 period, where minor discrepancies likely originate from a combination of differing treatment of the tropopause height and dataset masking, as well as differing effective radius values, with CMIP7's values unrealistic owing to the lack of multiwavelength information in GloSSAC for that period (section 2.2). However, the most important discrepancies are before 1979, the onset of the satellite era. There are numerous instances in which only one dataset exhibits a SAOD perturbation, including for perturbations larger than 0.02. Overall, the CMIP7 dataset contains more eruptions for the 1850-1960 period, whereas the CMIP6 dataset contains more for 1960-1980. This results in a global mean SAOD of 0.0135 for CMIP7 piControl climatology, 26% higher than the 0.0107 value for CMIP6. Beyond eruption occurrence, disagreements also exist on the magnitude of SAOD perturbations, with the CMIP7 dataset having notably higher perturbations for the early 1860's unidentified eruption (attributed to Kie Besi 1861 in CMIP7, Aubry et al., 2025), a cluster of eruptions injecting 0.6-3 Tg SO<sub>2</sub> in 1885-1886, the Mt Tarawera 1886 eruption or the Ksudach 1907 eruption. CMIP6 has greater perturbations for the Suwanosejima 1889 and Santa Maria 1902 eruptions. Furthermore, the latitudinal distribution of SAOD can also disagree between the two datasets, even for eruptions they have in common (Figure 11.b-i), e.g. in the early 1860s, or late 1890s. Hemispherically uniform perturbations in CMIP6, caused by scaling of single-station pyrheliometer measurements to entire hemispheres (Figure 1), are absent in CMIP7. Last, CMIP7 also includes the 1750-1849 period (Figure 11.a) to facilitate the running of historical simulations starting in 1750. This period is characterized by much stronger forcing than the "standard" CMIP historical period (1850-present day), with eruptions of Kie Besi (1760), Laki (1783-1784), an unidentified volcano (1809), Tambora (1815), Zavaritzki (1831) and Cosiguina (1835) ranging from 10-56 Tg SO<sub>2</sub> and totaling 190 Tg SO<sub>2</sub> among them.

Figure 10: Global mean SAOD at 550 nm for CMIP6 and CMIP7, from 1850 onwards. Thin lines show the piControl climatology, obtained from the 1850-2014 and 1850-2021 climatology for CMIP6 and CMIP7, respectively.

Figure 11: Latitudinal distribution of SAOD at 550 nm (logarithmic scale) for CMIP7 (left, panels a, b, d, f and h) and CMIP6 (right, panel c, e, g and i). For CMIP7, the top panel (a) shows 1750-1850 which is not included in CMIP6.

We acknowledge that the differences between the CMIP6 and CMIP7 datasets will motivate questions regarding which dataset is better, especially for the pre-satellite era. When it comes to the magnitude and spatial distribution of the forcing at any specific time, both datasets are highly uncertain. Volcanic SO<sub>2</sub> emission parameters used in both CMIP6 (for 7 eruptions) and CMIP7 are uncertain (e.g., Verkerk et al., 2025), although the CMIP7 emission inventory (Aubry et al., 2025) relies on source datasets that better reflect the state of the art in the ice-core and volcanological communities (Sigl et al., 2015; Toohey and

Sigl., 2017; Aubry et al., 2021b; Fang et al., 2023; Global Volcanism Program, 2025) compared to CMIP6, which used Gao 625 et al. (2008) as primary source for emissions. In particular, the CMIP7 approach enables us to make progress on the strong bias in the representation of small-to-moderate magnitude eruptions in CMIP6, albeit this is achieved by relying on an ice-core characterized by very high uncertainties (Fang et al., 2023) complemented by the geological record (Global Volcanism Program, 2025), which requires ad-hoc assumptions for SO<sub>2</sub> masses. CMIP6 used a more sophisticated aerosol model (AER2D, Arfeuille et al., 2014) than CMIP7 (EVA H v2, section 3) to translate emissions into stratospheric aerosol optical properties. 630 However, given large uncertainties in state-of-the-art interactive stratospheric aerosol models (Clyne et al., 2021; Quaglia et al., 2023), the use of a more sophisticated model does not imply greater accuracy. The provision of a volcanic SO<sub>2</sub> emission dataset for CMIP7 (Figure 3 and Aubry et al., 2025) will ultimately foster the development of interactive stratospheric aerosol models, and in turn inform our understanding of stratospheric aerosol forcing and climate impacts. Last, unlike CMIP7, CMIP6 635 attempted to make use of observational pyrheliometer data to constrain pre-satellite era forcing (Figure 1). However, the scarcity of measurements, the reliance on single stations (for most months) to derive global-scale aerosol optical properties, and the conversion of pyrheliometer measurement into SAOD are also subject to high uncertainties. Overall, we thus recommend to not consider either dataset superior to the other for any specific month or year, and highlight the need for further evaluation.

#### 5.2 Potential implications for simulating historical climate in CMIP7

The changes in mid-visible SAOD in CMIP7 relative to CMIP6 might have implications for simulating the historical climate in CMIP7. To provide an initial assessment, we estimate the global annual mean top-of-the-atmosphere effective radiative forcing (ERF) associated with each forcing dataset using the scaling from Marshall et al. (2020):

$$ERF = -20.7 (1 - e^{-\Delta SAOD})$$
 (5)

where ΔSAOD is the global annual mean 550 nm SAOD anomaly, calculated with respect to the minimum historical value. We then simulate global mean surface temperature using FaIR v2.1.4 (Smith et al. 2018, Leach et al. 2021), a reduced-complexity climate model calibrated using emissions from the IPCC Sixth Assessment Report (calibration 1.4.1, Smith et al., 2024). We force FaIR with historical CMIP6 greenhouse gases, ozone and aerosol precursor emissions and solar and land use forcings from the RCMIP dataset (Nicholls et al., 2020). Forcings linked to aerosol-radiation interaction, aerosol-cloud interaction, ozone, light absorbing particles on snow and ice, and stratospheric water vapor are calculated by the model based on emissions. We run two ensembles of simulations for the CMIP6 and CMIP7 volcanic forcing time series, where volcanic forcing is expressed in FaIR as an anomaly with respect to the pre-industrial climatology, i.e. 1850-2014 mean and 1850-2021 mean for the CMIP6 and CMIP7, respectively.

Consistent with SAOD (Figure 10), both the stratospheric aerosol radiative forcing (Figure 12.a) and simulated global mean surface temperature (Figure 12.b) are lower in CMIP7 for 1850-1900, with CMIP7 forcing being more negative by up to 1.5 W/m², and associated temperatures cooler by up to 0.2°C. The greatest change occurs after the unidentified 1860 eruption, but overall, over 1850-1900, the radiative forcing is typically more negative in CMIP7. Also consistent with SAOD changes, for 1963-1982, CMIP6 radiative forcing is more negative by up to 1 W/m², and associated temperatures cooler by up to 0.05°C.

Figure 12: Global mean top of the atmosphere (TOA) effective radiative forcing (ERF) (top, a) and surface air temperature (SAT) anomaly relative to pre-industrial (bottom, b) for CMIP6 and CMIP7, for 1850-2023.

Table 3 summarizes key metrics in terms of global mean SAOD, effective radiative forcing and surface air temperature associated with each dataset. Over 1850-2014 (the period common to CMIP6 and CMIP7), the CMIP7 mean SAOD (0.0138) and radiative forcing (-0.22 W/m²) are enhanced by 29% and 37% compared to CMIP6 (0.0107 SAOD, -0.16 W/m² forcing). Preindustrial control simulations, forced by the historical mean forcing, are cooler by 0.03°C in CMIP7 compared to CMIP6 (not shown). Last, relative to their respective pre-industrial baseline, simulations forced by the CMIP7 stratospheric aerosol dataset are 0.07°C cooler than CMIP6 over 1850-1900, but 0.03°C warmer over 2001-2014. Relative to 1850-1900, a key reference period for assessment reports by the Intergovernmental Panel on Climate Change (IPCC, 2021), simulated 2000-

2014 global mean surface temperatures are thus warmer by 0.1°C when using the CMIP7 stratospheric aerosol dataset instead of the CMIP6 one. Over 1960-1990, the period in which CMIP6 models are affected by a cold bias (e.g. Flynn and Mauritsen, 2020; Zhang et al., 2021), simulations forced by the CMIP7 dataset are 0.06°C warmer relative to pre-industrial (0.13°C warmer relative to 1850-1900) compared to simulations forced by the CMIP6 dataset. Our new dataset could thus affect tuning of CMIP7 climate models, and/or for their performance in reproducing historical temperature trends.

| Global mean metric                                 | CMIP6   | CMIP7   |
|----------------------------------------------------|---------|---------|
| 1850-2014 mean SAOD at 550 nm                      | 0.0107* | 0.0138  |
| 1850-2021 mean SAOD at 550 nm                      | NA      | 0.0135* |
| 1850-2014 mean TOA ERF (W/m <sup>2</sup> )         | -0.16   | -0.22   |
| 1850-1900 mean SAT relative to pre-industrial (°C) | 0.06    | -0.01   |
| 1960-1990 mean SAT relative to pre-industrial (°C) | 0.23    | 0.29    |
| 2001-2014 mean SAT relative to pre-industrial (°C) | 0.89    | 0.92    |
| 2001-2014 mean SAT relative to 1850-1900 (°C)      | 0.83    | 0.93    |

Table 3: Global mean of key metrics (SAOD, TOA ERF and SAT anomaly) for selected time period, for CMIP6 and CMIP7. The asterisks (\*) highlight the global mean SAOD value characterizing the preindustrial climatology for each dataset.

# **5.3** Other stratospheric aerosol properties




To complement 550 nm SAOD comparison between CMIP6 and CMIP7 presented in section 5.1, Figure 13 compares the global stratospheric mean effective radius, surface area density, and H<sub>2</sub>SO<sub>4</sub> number density for CMIP6 and CMIP7. The CMIP7 global mean effective radius (0.132 μm) is 12% larger than that of CMIP6 (0.118 μm) over the common period (1850-2014). This difference is largely driven by the difference in non-volcanic background effective radius, which is 0.115 μm in CMIP7 with no trend (inconsistent with the trend in non-volcanic background SAOD), and exhibits a linear trend in CMIP6 (0.09 in 1850, reaching 0.115 in 1978). The datasets also exhibit large discrepancies over the satellite era, despite good agreement on peak effective radius for the El Chichón (1982) and Pinatubo (1991) eruptions. First, over 1980-2004, CMIP7 has larger effective radius with the notable exception of good agreement for the peak effective radius following the El Chichón (1982) and Pinatubo (1991) eruptions. Given that both datasets use GloSSAC for this time period, differences are driven by differences in methodologies to derive effective radius from multiwavelength extinction efficiency measurement. In particular, the REMAP algorithm (Jörimann et al., 2025) used in CMIP6 fits a single-mode log-normal size distribution with a variable geometric standard deviation to fit extinction efficiency measurements, whereas CMIP7 uses a bimodal lognormal distribution with fixed geometric standard deviation of 1.8 and 1.25 for smaller and larger size mode respectively (section 3.3). Second, over 2005-2017, the effective radius exhibits no variability in GloSSAC because this period only has constraint on 1020 nm


extinction efficiency, with 525nm scaled from it, resulting in constant extinction ratio and thus effective radius using our methodology. In contrast, the effective radius or the standard deviation was parameterized in CMIP6 for periods with single wavelength (Jörimann et al., 2025), resulting in a more realistic, time-varying effective radius.

The CMIP7 mean surface area density (1.31 µm<sup>2</sup> cm<sup>-3</sup>) is similar to the CMIP6 mean (1.30 µm<sup>2</sup> cm<sup>-3</sup>) over 1850-2014. This is caused by a ≈35% lower non-volcanic background surface area density in CMIP7 compared to CMIP6 (e.g. 0.65 μm<sup>2</sup> cm<sup>-3</sup> vs 1.06 µm<sup>2</sup> cm<sup>-3</sup> over 1998-2001) compensated by the increased number of small-to-moderate magnitude eruption pre-satellite era in CMIP7. The global 1850-2014 mean H<sub>2</sub>SO<sub>4</sub> number density in CMIP7 (8.0 x 10<sup>8</sup> molecules cm<sup>-3</sup>) is 60% higher than in CMIP6 (5.4 x 108 molecules cm<sup>-3</sup>), which is again largely driven by the increased number of eruptions pre-satellite era, with a good agreement between CMIP6 and CMIP7 over the satellite era. The enhanced H<sub>2</sub>SO<sub>4</sub> number density over 1850-1978 might influence stratospheric ozone forcing. Overall, for all three variables shown in Figure 13, peak values associated with the Chichón Agung (1963),Εl (1982)and Pinatubo (1991)eruptions reasonable agreement. are in

Figure 13: Global stratospheric mean (a) effective radius (in  $\mu m$ ), (b) surface area density (in  $\mu m^2$  cm<sup>-3</sup>), and (c) H<sub>2</sub>SO<sub>4</sub> number density (in molecules cm<sup>-3</sup>, log scale) for CMIP6 and CMIP7, for 1850-2023.




# 6 Stratospheric aerosol optical properties for CMIP: Challenges, opportunities and operationalization.

# 6.1 Key uncertainties and challenges in the CMIP7 dataset

This study focuses on documenting the CMIP7 stratospheric aerosol optical properties and preliminary comparison to CMIP6, especially for SAOD. However, a comprehensive evaluation of the CMIP7 dataset remains required, and such evaluation is currently being undertaken as part of the Fresh Eyes on CMIP Modern Forcing project (<a href="https://wcrp-cmip.org/cmip7-task-teams/fresh-eyes-on-cmip/">https://wcrp-cmip.org/cmip7-task-teams/fresh-eyes-on-cmip/</a>). It will include comparisons of climatology and background aerosols between CMIP6 and CMIP7, and evaluation of CMIP7 aerosol optical properties against independent observational measurements such as pyrheliometers, total lunar eclipses, and stellar extinction (Sato al., 1993; Stothers, 1996; 2001), and satellite datasets such as Scanning Imaging Absorption Spectrometer for Atmospheric Chartography (SCIAMACHY) and Climate Data Record of Stratospheric Aerosols (CREST) (Bovensmann et al., 1999; Pohl et al., 2024; Sofieva et al., 2024). This work will shed light on the main limitations and uncertainties of our dataset, and on differences in stratospheric aerosol forcing between forcing generations.

Despite the pending evaluation of our dataset against independent data, the major sources of uncertainties are well identified. Whilst the satellite-era aerosol optical properties are subject to uncertainties, in particular before the Pinatubo 1991 eruption, the much longer pre-satellite era part of the dataset is without doubt more uncertain, with volcanic contributions dominating uncertainties. Pre-satellite era volcanic SO<sub>2</sub> emission uncertainties, which are discussed in more details in the companion paper are highly uncertain whether in terms of how much SO<sub>2</sub> was injected, its emission height and in particular how much reached the stratosphere, and even its emission location (in particular latitude) and date. Despite our best efforts to correct biases in small-to-moderate magnitude eruptions (injecting less than 3 Tg SO<sub>2</sub>) in CMIP7, it is also clear that many such eruptions are missing in the pre-satellite era. These emission uncertainties are compounded by volcanic aerosol modelling uncertainties, with simulated global mean SAOD differing by over a factor of 2 among different interactive stratospheric aerosol models (Clyne et al., 2021). Our use of a reduced-complexity aerosol model (EVA H v2, section 3), calibrated against the CMIP7 satellite-based volcanic SO<sub>2</sub> emission and stratospheric aerosol optical properties datasets, ensures consistency of the model output with recent observations. However, these are subject to significant uncertainties too. For the 1991 eruption of Mt Pinatubo, the SO<sub>2</sub> mass is 15 Tg SO<sub>2</sub> in our emission dataset (Aubry et al., 2025). However, observational estimates of SO<sub>2</sub> mass reach up to 22 Tg SO<sub>2</sub> (Guo et al., 2004), whereas a recent model inversion suggest that only 10.4 Tg SO<sub>2</sub> was injected into the stratosphere (Ukhov et al., 2023), with other modelling studies suggesting that an "effective" mass of 10 Tg SO2 is justified by the fact that a large fraction of SO<sub>2</sub> was scavenged rapidly by ash and ice in the volcanic cloud (e.g. Mills et al., 2016; Dhomse et al., 2020; Qaglia et al., 2023). This range of plausible SO<sub>2</sub> mass for the eruption with the most weight in the EVA H v2 calibration translates into a 50% uncertainty for 525 nm extinction and SAOD pre-satellite era in the CMIP7 aerosol optical property dataset, without even accounting for emission uncertainties of pre-satellite era eruptions. This is illustrated by the change in aerosol mass-SAOD scaling factor (Table 1) from 0.0187 to 0.0229 (Tg S)-1 from v1 to v2 of





EVA\_H, largely driven by a corresponding update of the Pinatubo SO<sub>2</sub> mass from 18 to 15 Tg SO<sub>2</sub> in MSVOLSO2L4. The combined changes in model parameterization (Equation 1), calibration method (Section 3.2) and updates and extension of calibration datasets also resulted in a change in EVA\_H's lowermost tropical aerosol loss timescale from 16.1 months in v1 (i.e. relatively moderate aerosol loss in the tropics given the 10.7 months latitudinal transport timescale and 2.3 months extratropical lowermost stratosphere loss timescale) to 57.7 months in v2 (i.e. virtually no aerosol loss in the tropics). This further illustrates the large volcanic aerosol modelling uncertainties, even when using a reduced-complexity, empirical model.

Beyond pre-satellite volcanic uncertainties, the pre-satellite non-volcanic stratospheric aerosols, which contribute 20% of the global mean 550 nm piControl SAOD (i.e. the 1850-2021 historical average), is a highly uncertain component of our dataset. First, defining a non-volcanic background from satellite-era measurements is challenging (Figure 6.a). Second, assessing how this background has evolved throughout the historical period is difficult. Historical simulations from multiple interactive stratospheric aerosol models with no volcanic emission would support this, with UKESM run suggesting that the linear trend imposed in CMIP7 is inadequate (Figure 6.c). Third, Scenario and piControl climatologies are currently equal and constant, even though the non-volcanic background stratospheric aerosol levels are expected to be different (Figure 6.c), and evolving in time for Scenario simulations with the evolution dependent on the underlying shared socio-economical pathway (SSP) considered (Figure 9). Fourth, developing datasets that isolate the volcanic and non-volcanic contributions to stratospheric aerosols should be considered in the future. For example, some models generate stratospheric aerosol interactively but choose to run historical simulations with prescribed stratospheric aerosol optical properties (and zeroed volcanic emissions) to ensure a stratospheric aerosol forcing consistent among simulations, and with other CMIP models. However, unless the radiative transfer models within these climate models ignore the optical properties of non-volcanic stratospheric aerosols simulated interactively, the contribution of non-volcanic aerosol would be double-counted when imposing the CMIP7 (or CMIP6) stratospheric aerosol optical properties. As another example, existence of a single dataset accounting for both volcanic forcing and a non-volcanic background influenced by anthropogenic aerosol emission biases the hist-nat and hist-aer experiments of the Detection and Attribution MIP (DAMIP, Gillet et al., 2016), which are part of the CMIP7 Assessment Fast Track (Dunne et al., 2025). Fifth, we currently do not account for any significant non-volcanic emission of stratospheric aerosol precursor pre-satellite era (e.g. pyroCb emissions, Kloss et al., 2019; Peterson et al., 2021, 2025) nor do we account for volcanic emission other than SO<sub>2</sub> (e.g. water, Legrande et al., 2016; Millán et al., 2022).

# **6.2** Opportunities for future improvements

Key future directions to improve the volcanic SO<sub>2</sub> emissions dataset, and in turn the stratospheric aerosol optical properties dataset, are discussed in detail in a companion paper on the emission dataset (Aubry et al., 2025). Three selected highlights are:





- 1) Enhance efforts to have a multidisciplinary perspective on emissions by incorporating together ice-core, remote-sensing and geological datasets. This was key in CMIP7 to better capture small-to-moderate magnitude eruptions underrepresented in ice-core datasets using the GVP database, as well as to constrain emission heights. For the latter, the use of the Independent Volcanic Eruption Source Parameter Archive (IVESPA, Aubry et al., 2021b) curated by the International Association of Volcanology and Chemistry of the Earth Interior (IAVCEI) enabled us to: i) use well-constrained values for SO<sub>2</sub> injection height for numerous events from 1900; ii) through IVESPA-derived relationships (Aubry et al., 2023), correct heights derived from isopleths, the most common method to estimate plume height from deposit, but which typically overestimates SO<sub>2</sub> injection height by 50%.
- Analyze more ice cores, and at high resolution. The datasets we use incorporate a relatively low number of ice cores, in particular in Greenland with 1 (Fang et al., 2023) to 3 (Toohey and Sigl, 2017) used over 1750-1900. There is a strong potential to produce a continuous historical volcanic SO<sub>2</sub> record using a larger number of cores (e.g. Gao et al., 2007; Crowley and Unterman, 2013; Gabriel et al., 2024). The use of the high-resolution D4i ice-core (Fang et al., 2023) was key in CMIP7 to better capture small-to-moderate magnitude eruptions and have direct constraints on their SO<sub>2</sub> mass, which is not the case with geological datasets (e.g. GVP). However, the fact that the D4i dataset is based on a single core means it is characterized by very high uncertainties, and does not provide any constraint on eruption latitude as deposition information is limited to Greenland.
  - 3) Incorporate existing information from ice-core isotope analyses, and push for new analyses focused on the historical period. Analysis of sulfur isotopes in ice cores provides information about the height of the sulfate aerosol perturbation (e.g. Burke et al., 2019), which would be a crucial addition to our pre-satellite era dataset. These analyses suggest that even for large-magnitude eruptions, a large fraction of sulfate deposited in ice cores might originate from the troposphere (Lee et al., 2024). For small-to-moderate magnitude eruptions, e.g. those constrained from D4i, we expect that this fraction might be even higher.

Future versions of the CMIP stratospheric aerosol dataset could make an important step forward for the pre-satellite era by combining the approaches used for CMIP7, i.e. entirely deriving the dataset from emissions, and CMIP6, i.e. relying as much as possible on pyrheliometer measurements, which are characterized by high uncertainties (section 1) but are the main observational source pre-satellite era. Should this approach be followed, it will be crucial to not let aerosol optical property measurements affect future version of the emission dataset, but instead to derive the aerosol optical dataset from a modified version of the emission dataset, with modifications rigorously documented and aimed at better matching the pyrheliometer record. Beyond pyrheliometer data, future dataset versions could consider in-situ measurements coming out from the SSiRC "data rescue of stratospheric aerosol measurements" effort (https://www.sparc-ssirc.org/data/datarescueactivity.html; Antuña-Marrero et al., 2020, 2021 and 2024, Dhomse et al., 2020), and indirect evidence from lunar eclipses (Guillet et al., 2024). In addition to data, advancing interactive stratospheric aerosol models and reducing uncertainties among them (e.g. Clyne et al., 2021) is a key priority as it will in turn improve estimates of stratospheric aerosol optical properties from emissions. Future

versions of our dataset are likely to continue to use a reduced-complexity volcanic aerosol model such as EVA\_H, because the trivial computational cost is key to facilitate the exploring of new developments and to operationalizing dataset production (see next section). However, reduced-complexity models inherently rely on full-blown interactive stratospheric aerosol models, as is the case for the parameterization of the sulfate burden-SAOD scaling used in EVA\_H (Aubry et al., 2020). For the satellite-era part of our dataset, pushing for a continued record of multi-wavelength measurements of stratospheric aerosol optical properties is critical to extend the dataset and to be able to use future stratospheric aerosol injection events, such as volcanic eruptions and pyrocumulonimbus, to improve our understanding of stratospheric aerosol processes. The use of existing multispectral satellite observations in the 2005-2017 period (Khanal et al., 2024), for which GloSSAC currently relies on single wavelength measurements, may help resolve biases in effective radius and aerosol properties in the current version of the dataset (Figure 13).

Beyond stratospheric aerosols, bespoke forcing datasets might need to be developed to account for volcanic water and halogen injection forcing. The former was made particularly important by the Hunga 2022 eruption which injected 146 Tg of water into the stratosphere (Millán et al., 2022) and will have contributed a small annual-decadal scale greenhouse forcing (e.g. Sellitto et al., 2022; Jenkins et al., 2023), even though the associated surface warming might be dominated by the surface cooling associated with the stratospheric aerosol forcing (e.g., Schoeberl et al., 2024; Stenchikov et al., 2025). Furthermore, as the CMIP Climate Forcing Task Team currently does not include a group dedicated to tropospheric aerosol forcing, modelling groups have no recommendations to implement associated emissions or forcing. Jongebloed et al. (2023) suggests that tropospheric volcanic sulfur emission inventories commonly used by climate models underestimate pre-industrial emissions, which would lead to lead to significant underestimation of tropospheric volcanic aerosol forcing, and in turn overestimation of anthropogenic aerosol forcing (Schmidt et al., 2012; Carslaw et al., 2013). Recent satellite emission inventories (e.g. Carn et al., 2017) also show significant decadal scale variability in tropospheric emissions, which is currently ignored in CMIP.

#### 6.3 Requirements for dataset operationalization

Naik et al. (2025) highlight the critical need to operationalize climate forcing production, i.e. move from the production of standardized climate forcing datasets synchronized with CMIP cycles, every 5-7 years, to annual production. To reach this goal for stratospheric aerosol optical properties, we review below key requirements and vulnerabilities. Our dataset relies on the existence of the following key resources:

- i. GloSSAC (Thomason et al., 2018; Kovilakam et al., 2020), which to date is the only multiwavelength stratospheric aerosol optical property dataset covering 1979-present day.
- ii. Version 2 of the EVA\_H reduced-complexity aerosol model (Aubry et al., 2020), an extension of the Easy Volcanic Aerosol originally developed by Toohey et al. (2016).
- iii. MSVOLSO2L4 (Carn, 2022), which to date is the only volcanic emission inventory covering 1979-present day.



- iv. The eVolv2k (Toohey and Sigl, 2017), Sigl et al. (2015), and D4i (Fang et al., 2023) ice core records of volcanic emissions
- v. The Global Volcanism Program Volcanoes of the World database (Global Volcanism Program, 2025)
- vi. Publications from the volcanology community documenting eruption source parameters, and the IVESPA eruption source parameter database (Aubry et al., 2021b)
  - vii. Publications from the volcanology and ice-core communities fingerprinting sulfate deposition peak in ice-core to specific eruptions (e.g. Plunkett et al., 2023).
  - viii. Person time to produce historical, pre-industrial and Scenario datasets from the resources listed above.
- Data from sources iii-vii were used to build our stratospheric volcanic SO<sub>2</sub> emission dataset (Aubry et al., 2025, briefly described in section 2.1) on which the stratospheric aerosol optical property dataset rely for the pre-satellite era (1750-1978). Furthermore, the MSVOLSO2L4 dataset was key to calibrating EVA\_H version 2, and thus for delivering the pre-satellite era aerosol optical property dataset.
- Among the listed resources, the operationalization of stratospheric aerosol optical property provision is most vulnerable to the continued provision (including extensions) of the GloSSAC dataset. This provision itself is vulnerable to the continued availability of multiwavelength measurements of stratospheric aerosol optical properties (Salawitch et al., 2025), in particular to potential risk of discontinuity in reliable solar occultation records if SAGE III/ISS operations end. Discontinuation of GloSSAC would incur significant additional workload a barrier to operationalization to use multiple stratospheric aerosol optical property records and merge them as consistently as possible, or to change method to produce satellite-era optical properties, e.g. by deriving them from emission as for pre-satellite era.

The second most important resource for operationalization might be viii, i.e. the person time required to produce CMIP datasets from a collection of resources (i-vii). The support of the CMIP International Project Office and European Space Agency was critical to produce this dataset for CMIP7. In the future, even for simple dataset extensions, i.e. with no change to periods included in previous dataset period, regular time investment is required to communicate with source dataset providers, produce the extension, ensure files comply with latest CMIP format requirements, and communicate with users. This workload is significantly enhanced for any update, i.e. changes to the period already covered by the previous version, and their documentation. Reasons for updates include incorporation of new versions of source datasets (iii-vii) and models (ii), incorporation of new source datasets or models in the dataset production workflow (Figure 2), or major changes to the workflow used to produce the dataset (as was done going from CMIP6 to CMIP7). To facilitate this workload and make it more resilient to future changes in the team producing stratospheric aerosol optical property for the CMIP Climate Forcing Task Team, all scripts used to produce our emission and stratospheric aerosol optical property datasets from resources i-vii above (https://doi.org/10.5281/zenodo.17295697) are available on zenodo and GitHub (https://github.com/thomasaubry/CMIP7 stratforcing v2.2.1) (see code availability statement).







Last, although vulnerabilities to the continued availability of resources ii-vii might be less critical for operationalization, our comparison of the CMIP6 and CMIP7 datasets and associated simulations using a reduced-complexity climate model (section 5) demonstrate that:

- Some of the most important advances for stratospheric aerosol optical properties might come from new development for the pre-satellite era source datasets and the stratospheric aerosol modelling communities.
- Such advances significantly affect historical (Figure 12, Table 3), pre-industrial and scenario (Chim et al., 2025) climate simulations.

The continued provision of best-effort stratospheric aerosol datasets to the climate modelling community thus relies on funding agencies to continue to support the work of the ice-core, satellite, field volcanology and stratospheric aerosol modelling communities, including but by no means limited to supporting data collection effort encapsulated in iii-vii, development of the EVA model family (ii), and the research avenues highlighted in section 6.2.

## 7 - Conclusions

This paper documents the stratospheric aerosol optical datasets (version 2.2.1) produced in support of phase 7 of the Coupled Model Intercomparison Project. The historical dataset covers 1750-2023 and, as in CMIP6 (Luo et al., 2018), relies on GloSSAC for the satellite era (1979 onwards). For the pre-satellite era, our methodology significantly differs from CMIP6, with volcanic perturbations to aerosol optical properties fully derived from volcanic sulfur emissions using version 2 EVA\_H, a reduced-complexity volcanic aerosol model. The volcanic stratospheric sulfur emission dataset, documented in a companion paper (Aubry et al., 2025), primarily relies on bipolar ice-core arrays pre-satellite era, but it is enhanced by the use of high-resolution Greenland core and geological databases used to better capture frequent small-to-moderate magnitude eruptions injecting less than 3 Tg SO<sub>2</sub>. Version 2 of EVA\_H includes a more realistic aerosol size distribution, an improved sensitivity of the magnitude and lifetime of forcing to SO<sub>2</sub> injection mass, latitude and height, and is calibrated against satellite-derived emission and optical properties for 1979-2023. Volcanic stratospheric aerosol optical property perturbations pre-satellite era were added to a background, non-volcanic climatology derived from the 1998-2001 volcanically quiescent period, with a trend over 1850-1978 accounting for increasing anthropogenic aerosols. A monthly stratospheric aerosol climatology is derived from the 1850-2021 average for both pre-industrial and Scenario (future) simulations, with a 10-year ramp over 2022-2031 for scenarios simulations to ensure a smooth transition from the historical period.

Because the historical pre-satellite optical properties are derived differently from CMIP6, the CMIP7 dataset differs in several ways. In particular, the representation of small-to-moderate magnitude eruptions is improved in CMIP7, yielding larger historical forcing. The 1850–2014 global mean 550 nm SAOD is 0.0138 in CMIP7 versus 0.0107 in CMIP6, representing an increase of 29%. The pre-industrial global annual mean 550 nm SAOD is 0.0135 in CMIP7 (derived from the 1850-2021 mean) versus 0.0107 in CMIP6 (derived from the 1850-2014 mean), representing an increase of 26%. In reduced-complexity



climate model simulations the global-mean surface temperature is colder by 0.07 °C in 1850–1900 using the CMIP7 forcing compared with using the CMIP6 forcing. Over 1960–1990, simulated temperature is 0.06 °C warmer when using the CMIP7 forcing which might partially alleviate the cold bias in that period in CMIP6 models. Despite improvements in small-to-moderate magnitude eruption representation and the use of more recent ice-core emission inventories in CMIP7 (Sigl et al., 2015; Toohey and Sigl, 2017; Fang et al., 2023) compared to CMIP6 (Gao et al., 2008), large uncertainties remain, especially for the pre-satellite era. We therefore advise against treating either dataset as uniquely superior for any specific month or year and highlight the need for further evaluation. We discuss future research avenues to improve the dataset, in particular improvements to the pre-satellite era emission dataset and volcanic aerosol model, the incorporation of indirect stratospheric aerosol optical property observations into the dataset, and addressing the limitation of GloSSAC to single-wavelength measurements over the 2005-2017 period. Finally, we outline requirements to operationalize the dataset production, i.e. update it on an annual basis instead of a CMIP cycle (5-7 years) basis, and stress the critical importance of continuing multi-wavelength stratospheric aerosol optical property measurements from space, as well as continued support to the CMIP Climate Forcing Task Team.

## **Code availability**

All codes used to produce the CMIP7 dataset and source datasets required to run them are available through Zenodo (Aubry, 2025, <a href="https://doi.org/10.5281/zenodo.17295697">https://doi.org/10.5281/zenodo.17295697</a>) and GitHub (<a href="https://github.com/thomasaubry/CMIP7">https://github.com/thomasaubry/CMIP7</a> stratforcing v2.2.1). Codes used to interpolate the files we provide on ESGF on any list of wavelength ranges inputted by the user are available at <a href="https://github.com/MetOffice/CMIP7">https://github.com/MetOffice/CMIP7</a> volcanic aerosol forcing/.

#### Data availability

All dataset produced for this paper are available on the Earth System Grid Federation at <a href="https://aims2.llnl.gov/search?project=input4MIPs&versionType=all&&activeFacets=%7B%22source\_id%22%3A%5B%22">https://aims2.llnl.gov/search?project=input4MIPs&versionType=all&&activeFacets=%7B%22source\_id%22%3A%5B%22</a>
<a href="https://aims2.llnl.gov/search?project=input4MIPs&versionType=all&&activeFacets=%7B%22source\_id%22%3A%5B%22">https://aims2.llnl.gov/search?project=input4MIPs&versionType=all&&activeFacets=%7B%22source\_id%22%3A%5B%22</a>
<a href="https://aims2.llnl.gov/search?project=input4MIPs&versionType=all&&activeFacets=%7B%22source\_id%22%3A%5B%22">https://aims2.llnl.gov/search?project=input4MIPs&versionType=all&&activeFacets=%7B%22source\_id%22%3A%5B%22</a>
<a href="https://aims2.llnl.gov/search?project=input4MIPs&versionType=all&&activeFacets=%7B%22source\_id%22%3A%5B%22">https://aims2.llnl.gov/search?project=input4MIPs&versionType=all&&activeFacets=%7B%22source\_id%22%3A%5B%22</a>
<a href="https://aims2.llnl.gov/search?project=input4MIPs&versionType=all&&activeFacets=%7B%22source\_id%22%3A%5B%22">https://aims2.llnl.gov/search?project=input4MIPs&versionType=all&&activeFacets=%7B%22source\_id%22%3A%5B%22</a>
<a href="https://aims2.llnl.gov/search?project=input4MIPs&versionType=all&&activeFacets=%7B%22source\_id%22%3A%5B%22">https://aims2.llnl.gov/search?project=input4MIPs&versionType=all&&activeFacets=%7B%22source\_id%22%3A%5B%22</a>
<a href="https://aims2.llnl.gov/search?project=input4MIPs&versionType=all&&activeFacets=%7B%22source\_id%22%3A%5B%22</a>
<a href="https://aims2.llnl.gov/search?project=input4MIPs&versionType=all&&activeFacets=%7B%22source\_id%22%3A%5B%22</a>
<a href="https://aims2.llnl.gov/search?project=input4MIPs&versionType=all&&activeFacets=%7B%22source\_id%22%3A%5B%22</a>
<a href="https://aims2.llnl.gov/search?project=input4MIPs&versionType=all&&activeFacets=%7B%22source\_id%22%3A%5B%22</a>

ZN produced the code to make the dataset comply with CMIP data standards. TJA wrote the original draft, and all authors contributed to review and editing. TJA and AS acquired funding.

## **Competing interests**

VN is a topic editor of this special issue. The authors have no other competing interests to declare.

#### Special issue statement

This manuscript is submitted to the special issue "Coupled Model Intercomparison Project Phase 7 (CMIP7) forcings and inputs – development, documentation, and evaluation".

## 935 Acknowledgements

We warmly thank the numerous people who gave feedback on the dataset during feedback and production, including the VolImpact 2025 workshop attendees, the "Perspectives on Stratospheric Aerosol Observations" team of the International Space Science Institute, Paul Durack, Claudia Timmreck, Claire Macintosh, Jing Feng, Martine Michou, Makoto Deushi, Gareth Jones, Anton Laakso, Jaehee Lee, and many others who contributed through CMIP meetings and workshops or informal discussions with us. TJA and IS thank Jane Mulcahy, Colin Jones, and Jeremy Walton for their support in producing the UKESM simulation show in figure 6.

# Financial support


TJA acknowledges generous support from the CMIP International Project Office (project 4000136906), the European Space Agency "Volcanic forcing for CMIP" project of the Climate Change Initiative (CCI) (ESA Contract No. 4000145911/24/I-LR), the University of Exeter (through the Camborne School of Mines and the Department of Earth and Environmental Sciences research funds), and a travel award from the Canada-UK foundation. MT and SK acknowledge support from the Canadian Space Agency (grant no. 21SUASOCSA). MMC is supported by the Croucher Foundation through the Croucher Postdoctoral Fellowship. EZ acknowledges funding through the Deutsche Forschungsgemeinschaft (STACY, grant no. 395588486) and the Federal Ministry of Education and Research of Germany (PalMod III, grant no. 01LP2311C). Michael Sigl received funding from the European Research Council under the European Union's Horizon 2020 research and innovation programme (grant number: 820047) and from the Swiss State Secretariat for Education, Research and Innovation (SERI) under

the contract no. MB22.00030. Zebedee Nicholls received funding from the European Space Agency (ESA) as part of the GHG Forcing For CMIP project of the Climate Change Initiative (CCI) (ESA Contract No. 4000146681/24/I-LR-cl) and the European Union's Horizon 2020 research and innovation program (grant agreement no. 101003536) (ESM2025).

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
