# Peer review of "Stratospheric aerosol forcing for CMIP7 (part 1): Optical properties for pre-industrial, historical, and scenario simulations (version 2.2.1)"

_EGUsphere, 2025_

## Author Comment (AC1)

We thank the reviewer their supportive comments on our work and their suggestions which will help us improve the manuscript. We briefly respond below to any point that required a response. We did not respond to any other technical comments and minor comments that are editorial in nature, but the helpful suggestions will be incorporated in full in our revised manuscript.

***Specific comments:***

*Line 75; Section 1.2:*

*It is correct that there is no complete published CMIP6 dataset description, however, in the supplement to Jörimann et al. (2025) the SAGE-3λ record is documented, which was the latest iteration that went directly into the CMIP6 forcings. Using this source, you can confirm that for 1961-1978 pyrheliometer data from Stothers (2001) - which you cite - are used. Please also check in this entire subsection and Fig. 1a, if the supplement can complement your overview of the CMIP6 forcing. Since SAGE-3λ uses three wavelengths (wherever possible), it should also be specified that GloSSAC provides more than two wavelengths (line 83) on occasion.*

We thank the reviewer for pointing to how we could better use Jörimann et al. (2025). We will implement suggestions on GloSSAC wavelengths and pyrheliometer use, and review whether we could improve this section considering Jörimann et al. (2025).

*Line 211:*

*You choose the VEI 4 SO$_2$ mass for GVP events to match the anomaly from 1998-2023 "... (defined **as** the deviation from its minimum) ...". I agree that the chosen time period is fit for this purpose, but it is not exactly clear what the deviation from the minimum is. Is the minimum the very lowest SAOD data point found in this time period? If so, is it representative for an "undisturbed" stratospheric aerosol or could it be an outlier? Or could an average such as 1999-2003 (quasi-quiescent state) be taken as the minimum to derive the deviation against? Please elaborate what is meant by the minimum and why it was chosen this way.*

The minimum indeed refers to the lowest value over the time period, and we believe the use of minimum is clear enough. However, to improve clarity and address the well justified reviewer comment, we will add a sentence quoting the value of this minimum and contrasting it to the value of the 1998-2001 mean SAOD. The relatively small difference in these values means that the definition of the baseline has little effect on our definition of VEI 4 default mass. We chose the actual minimum because there is no time period which is perfectly volcanically quiescent, and taking the minimum thus likely already represents a conservative estimate of the true baseline.

We acknowledge that this approach is crude and, beyond the question of how to define a baseline, is subject to limitations such as non-volcanic influences on SAOD over 1998-2023 or a varying anthropogenic contribution to SAOD (in particular through cross-tropopause aerosol transport).

*Line 232:*

*Periods without SAGE coverage are supplemented, but periods with SAGE coverage are also partly supplemented, especially at high latitudes, can you confirm this? The sentence could simply be extended similar to: "Periods without SAGE coverage **and high latitude data not***

*captured during SAGE coverage* are supplemented by complementary spaceborne and
ground-based observations ..."

This is an excellent suggestion and we will implement it.

*Lines 294-297:*

*I find this sentence hard to understand due to its length. Consider splitting it into two sentences for the reader's convenience.*

We will split in two sentences.

*Line 347:*

*It says that after the initial model parameter search with eruption masses **higher** than 0.1 Tg $SO_2$, the refined search then respects the other eruptions "... with an upper end of stratospheric $SO_2$ mass **higher** than 0.1 Tg $SO_2$." Is it not supposed to be "lower" here, instead of "higher" again? Otherwise I do not understand the distinction between the initial and refined search.*

The distinction is between best estimate and upper-end estimate. We will change this sentence
to:

"[...] with a best estimate of stratospheric SO2 mass higher than 0.1 Tg SO2. We then did a
refined search allowing for adjustment of SO2 masses for the 88 eruptions with an upper-end
estimate (as opposed to best estimate) of stratospheric SO2 mass higher than 0.1 Tg SO2."

*Line 423:*

*$M_{H2SO4}$ must be the molar weight, not the molar concentration of $H_2SO_4$. This follows from dimensional analysis of your Eq. (2), which yields mass per mole for $M_{H2SO4}$, not number per volume as it would be for molar concentration.*

Thanks for catching that mistake, we will correct it in the manuscript.

*Line 505:*

*From previous description it seems that the extinction coefficients are derived using version 2 (exclusively) of EVA_H. If so, specify "EVA_H v2", as the term "EVA_H" has been used before to distinctly talk about the EVA_H group (v1 and v2). The version number should also be used in the subplot titles of Fig. 7 & 8*

This is correct and we will specify v2 throughout the manuscript where required, including
where suggested by the reviewer.

*Line 526:*

*In this section the wavelengths, on which you produced ext, ssa, asy are given, however, the radiative transfer models operate with wavelength bands. In the data you provide both "wavelength" and "wavelength_bnds" variables and it seems that the wavelength is the (linear!) center or midpoint of each wavelength band. The wavelength bands are continuous across the spectrum. Do the data (e.g. ext) you report on the 39 wavelengths, correspond to the data you computed for the respective wavelength band? If so, can you make it clear in the text, notably in this section (4.3)? Also specify what exactly is meant by "properties at arbitrary wavelengths" on line 545 in this context.*

All data provided is for the corresponding wavelength, not wavelength band. However, the script briefly documented in section 4.5 calculates properties averaged over user-inputted wavelength bands. We will clarify that distinction in section 4.3. "arbitrary wavelengths" refers to the fact we can produce these properties at any wavelength other than 525 and 1020 nm, although in the CMIP7 framework, we produced them at the list of 39 wavelengths provided. We will also clarify this in the section.

*Lines 574-575:*

*The colors you mention (red, green, and black) do not correspond to what is shown in Fig. 9 (different shades of red). Change the caption or the lines in the plot.*

Thanks for catching that mistake, we will correct it in the manuscript.

*Line 1130:*

*Please add the ETH research collection item Luo (2017), which was created as a more accessible and persistent item that contains the CMIP6 data and some documentation and also has a DOI (https://doi.org/10.3929/ethz-b-000715155).*

*Since the specific file on the FTP file server that you reference in Luo (2018) is not in the ETH research collection item, I suggest that you either keep the FTP link as the separate citation you already have, or add a note to the Luo (2017) citation to mention that some description documents are only available there, similar to ("with additional information accessible at …").*

*Finally, update the last access.*

*Citations not already in the manuscript:*

*Luo, B.: SAGE-3λv4: Stratospheric aerosol data for use in CMIP6 models, ETH Research Collection [data set], https://doi.org/10.3929/ethz-b-000715155, previously distributed through ftp://iacftp.ethz.ch/pub_read/luo/CMIP6_SAD_radForcing_v4.0.0 (last access: 8 June 2025), 2017.*

Thank you, we will follow these suggestions in the revised manuscript.